# TRAF6 directs FOXP3 localization and facilitates regulatory T-cell function through K63-linked ubiquitination

Xuhao Ni[1,2,†], Wei Kou[3,†], Jian Gu[1,†], Ping Wei[2,3], Xiao Wu[1], Hao Peng[1], Jinhui Tao[2], Wei Yan[1], Xiaoping Yang[2], Andriana Lebid[2], Benjamin V Park[2], Zuojia Chen[4], Yizhu Tian[1], Juan Fu[2], Stephanie Newman[5], Xiaoming Wang[6] (ID), Hongbin Shen[7], Bin Li[4], Bruce R. Blazar[8], Xuehao Wang[1,9], Joseph Barbi[5,*] (ID), Fan Pan[2,**] (ID) & Ling Lu[1,9,***] (ID)

## Abstract

Regulatory T cells (Tregs) are crucial mediators of immune control. The characteristic gene expression and suppressive functions of Tregs depend considerably on the stable expression and activity of the transcription factor FOXP3. Transcriptional regulation of the Foxp3 gene has been studied in depth, but both the expression and function of this factor are also modulated at the protein level. However, the molecular players involved in post-translational FOXP3 regulation are just beginning to be elucidated. Here, we found that TRAF6-deficient Tregs were dysfunctional *in vivo*; mice with Treg-restricted deletion of TRAF6 were resistant to implanted tumors and displayed enhanced anti-tumor immunity. We further determined that FOXP3 undergoes K63-linked ubiquitination at lysine 262 mediated by the E3 ligase TRAF6. In the absence of TRAF6 activity or upon mutation of the ubiquitination site, FOXP3 displayed aberrant, perinuclear accumulation and disrupted regulatory function. Thus, K63-linked ubiquitination by TRAF6 ensures proper localization of FOXP3 and facilitates the transcription factor's gene-regulating activity in Tregs. These results implicate TRAF6 as a key posttranslational, Treg-stabilizing regulator that may be targeted in novel tolerance-breaking therapies.

**Keywords** cancer; FOXP3; TRAF6; Tregs; ubiquitin

**Subject Categories** Cancer; Immunology; Post-translational Modifications, Proteolysis & Proteomics
**The EMBO Journal (2019) 38: e99766**

## Introduction

In order for the immune system to function properly, the targets and amplitude of immune responses must be tightly controlled. Regulatory T cells (Tregs) are among the safeguards that prevent aberrant immune responses including those underlying autoimmunity, inflammatory diseases, or allergy (Sakaguchi *et al*, 2008). This critical function of Tregs is mediated by several suppressive mechanisms including production of anti-inflammatory cytokines (IL-10, TGFβ, IL-35, etc.), expression of co-inhibitory receptors and ligands, and the disruption of effector cell division and metabolism (Vignali *et al*, 2008). The expansion or enhancement of Tregs may be an effective means to enforce immune tolerance to transplanted grafts or to restore immune homeostasis in patients suffering from auto-immune pathologies (Wang *et al*, 2011). Conversely, Treg depletion or inhibition approaches can be exploited to provoke effective anti-tumor immunity in the cancer setting. Therefore, a thorough understanding of the molecules and pathways important for the function

1   Translational Medicine Research Center of Affiliated Jiangning Hospital, Liver Transplantation Center of First Affiliated Hospital, and Collaborative Innovation Center for Cancer Medicine, Nanjing Medical University, Nanjing, Jiangsu, China
2   Immunology and Hematopoiesis Division, Department of Oncology, Sidney Kimmel Comprehensive Cancer Center, Johns Hopkins University School of Medicine, Baltimore, MD, USA
3   Department of Otolaryngology, Pediatric Research Institute, The Children's Hospital of Chongqing Medical University, Chongqing, China
4   Shanghai Institute of Immunology, Shanghai JiaoTong University School of Medicine, Shanghai, China
5   Department of Immunology, Roswell Park Comprehensive Cancer Center, Buffalo, NY, USA
6   State Key Laboratory of Reproductive Medicine, Department of Immunology, Nanjing Medical University, Nanjing, China
7   Department of Epidemiology and Biostatistics, School of Public Health, Collaborative Innovation Center for Cancer Medicine, Nanjing Medical University, Nanjing, Jiangsu, China
8   Division of Blood and Marrow Transplantation, Department of Pediatrics, University of Minnesota, Minneapolis, MN, USA
9   State Key Laboratory of Reproductive Medicine, Nanjing Medical University, Nanjing, Jiangsu, China
    *Corresponding author. Tel: +1 7168451189; E-mail: joseph.barbi@roswellpark.org
    **Corresponding author. Tel: +1 4432877264; E-mail: fpan1@jhmi.edu
    ***Corresponding author. Tel: +86 025 68033934; E-mail: lvling@njmu.edu.cn
    † These authors contributed equally to this work

of these critical suppressor cells can have broad therapeutic implications.

The suppressive capabilities of Tregs are underpinned by a characteristic gene expression profile defined in large part by the transcription factor FOXP3. This forkhead/winged-helix family member works in concert with multiple co-regulator molecules (Eos, IRF-4, etc.) and a Treg-specific array of epigenetic modifications to shape the transcriptional landscape of Tregs (Pan *et al*, 2009; Fu *et al*, 2012; Lu *et al*, 2014, 2017; Morikawa & Sakaguchi, 2014). This is typified by the stabilized expression of Treg-associated genes (e.g., *Ctla4*, *Il2ra*) and the silencing of genes associated with effector T-cell lineages (e.g., *Il2*, *Ifng*). Stable expression of FOXP3 is not only characteristic of Tregs, but it is also key for their identity and suppressive function.

Indeed, the mechanisms controlling *Foxp3* transcription are important for both the generation of Tregs and the maintenance of their suppressive phenotype (Josefowicz *et al*, 2012). Accordingly, these mechanisms have been extensively studied. Mechanisms of protein-level regulation of FOXP3 expression and function, on the other hand, have only recently gained appreciation. Nevertheless, a growing body of evidence shows that the stability and activity of FOXP3 (and hence Tregs) can be significantly influenced by this layer of regulation. Multiple examples of posttranslational modifications to FOXP3 have been shown to have significant consequences for Treg function. For example, FOXP3 acetylation at specific lysine residues improves the stability of FOXP3 expression and enhances the association of the transcription factor with its target genes (Tao *et al*, 2007; van Loosdregt *et al*, 2010; Nie *et al*, 2013). Also, phosphorylation of FOXP3 at distinct sites has been found to have positive and negative effects on both its activity and Treg functions (Morawski *et al*, 2013; Deng *et al*, 2015).

We and others have documented how ubiquitination affects the FOXP3 protein pool and the suppressive capacities of Tregs (Chen *et al*, 2013; van Loosdregt *et al*, 2013). Protein ubiquitination events regulate diverse biological functions that can depend on both the specific target residue on the modified protein and the nature of the ubiquitin-to-ubiquitin linkages within the added polymer chains. For instance, the addition of ubiquitin monomers linked together at lysine residue 48 (K48-linkage) often marks a target protein for degradation by the 26S proteasome (Pickart, 1997). Other types of ubiquitination, such as those involving K63-linked ubiquitin molecules, can be important for the activation of signaling molecules and protein trafficking events (Kirisako *et al*, 2006; Wang *et al*, 2012). Previously, we and our colleagues found that FOXP3 is subject to K48-linked polyubiquitination that mediates the transcription factor's degradation via the proteasome. Therapeutically invoking or preventing this modification could effectively sabotage (Chen *et al*, 2013) or stabilize Treg suppressive function (van Loosdregt *et al*, 2013), respectively. However, the potentially diverse outcomes of different FOXP3 ubiquitination events and the molecular pathways involved remain incompletely understood.

Here, we report a distinct pathway of K63-type FOXP3 ubiquitination capable of promoting the proper nuclear localization and regulatory activity of FOXP3 as well as the phenotypic stability and suppressive function of Tregs. This pathway was found to be mediated by the TNF receptor-associated factor protein family member, TRAF6. It is known that mice lacking TRAF6 generally, in T cells, and specifically in FOXP3+ Tregs display compromised immune control and robust T-cell activation at baseline compared to wild-type controls (Pickart, 1997; Chiffoleau *et al*, 2003; King *et al*, 2006; Kirisako *et al*, 2006; Shimo *et al*, 2011; Metzger *et al*, 2012; Wang *et al*, 2012; Muto *et al*, 2013). However, the mechanisms responsible for the regulation of FOXP3 by TRAF6 remain to be fully explored. Here, we report that in addition to baseline defects in the enforcement of immune tolerance, Treg-specific TRAF6 deficiency leads to impaired Treg function *in vivo*, heightened anti-tumor immunity, and resistance to implanted tumors. The important role for TRAF6 in Tregs underscored by these observations was revealed to depend upon the E3 ligase's ability to interact with FOXP3 and execute K63-type ubiquitination at a specific lysine residue (K262) on the transcription factor. Furthermore, we show that the zinc finger and leucine zipper domains of FOXP3 are critical for TRAF6 interaction and the process of K63 ubiquitination. Importantly, we demonstrate that in TRAF6-deficient Tregs the FOXP3 is prone to aberrant accumulation outside the nucleus, impaired regulatory activity, and heightened protein instability relative to wild-type controls. Taken together, these results demonstrate a hitherto unknown, posttranslational mechanism controlling both the regulatory activity of FOXP3 and Treg suppressive potency. They also identify TRAF6 as a potential target for focused, tolerance-breaking immunotherapies.

# Results

## TRAF6 is critical for enforcement of immune homeostasis by Tregs

TRAF6 plays a potentially significant role in Treg biology. In line with this notion, high levels of TRAF6 transcript were found in naïve CD4+ T cells differentiating into Tregs *in vitro* (induced or iTregs), but not those committing to other T helper lineages (Fig 1A and Appendix Fig S1A). We further found that naturally occurring Tregs (nTregs) freshly isolated from the peripheral blood of healthy human donors expressed TRAF6 mRNA to a greater degree than their non-Treg CD4+ counterparts (Fig 1B and Appendix Fig S1B). This preferential expression of TRAF6 by multiple Treg subsets further implicated the E3 ligase as a key factor in the development and biology of these important suppressor cells.

To further investigate the importance of TRAF6 for Treg identity and function, we explored the consequences of TRAF6 deficiency *in vivo*. Previously, global TRAF6 knockout was found to precipitate multi-organ autoimmunity (Shimo *et al*, 2011). Similarly, T-cell-specific deletion of TRAF6 results in lymphoproliferative disease and systemic immune activation tied to T cells resistant to the suppressive function of Tregs (King *et al*, 2006). We set out to determine how much of this immune dysregulation stems from the absence of TRAF6 activity in Tregs by observing the consequences of *Foxp3*-driven TRAF6 deletion. Here, mice possessing a TRAF6 gene flanked with LoxP sites were crossed to mice expressing the Cre-recombinase under the control of the *Foxp3* promoter. The resulting Treg-specific knockouts (Traf6$^{fl/fl}$Foxp3Cre$^+$) and their wild-type littermates (Traf6$^{fl/fl}$Foxp3Cre$^-$, "WT") were monitored for indications of disrupted immune control. Indeed, Traf6$^{fl/fl}$Foxp3Cre$^+$ mice displayed signs of lymphoproliferative

disease. The lymph nodes and spleens of these mice were noticeably enlarged relative to the tissues of their wild-type littermates (Fig 1C). Increased cellularity was also noted in these lymphoid tissues in the absence of Treg-derived TRAF6 expression (Fig 1D). Flow cytometry analysis of lymphocyte surface markers revealed that both the $CD4^+$ and $CD8^+$ T-cell compartments of $Traf6^{fl/fl}Foxp3Cre^+$ mice harbored greater proportion of cells displaying an activated surface marker profile ($CD44^{high}/CD62L^{low}$) and fewer resting/naïve ($CD44^{low}/CD62L^{high}$) cells, indicative of enhanced baseline immune activation (Fig 1E, H, F and I). Furthermore, the frequencies of cells producing proinflammatory cytokines (IFN-$\gamma$, IL-17) were noticeably increased in the lymph nodes and spleens of $Traf6^{fl/fl}Foxp3Cre^+$ mice relative to wild-type controls at baseline (Appendix Fig S1C and D). Commensurate with these indications of poorly enforced immune tolerance and a propensity toward T-cell activation, $Traf6^{fl/fl}Foxp3Cre^+$ mice display stunted weight gain with age compared to their wild-type littermates (data not shown)—an observation in line with another recent study (Muto et al, 2013).

These observations clearly illustrate the importance of TRAF6 in the broad maintenance of immune tolerance by Tregs. As stable and robust FOXP3 expression is central to the suppressive phenotype of Tregs, a positive role for TRAF6 in the activation or maintenance of this factor could explain the effects of knocking out this enzyme. Indeed, prior work has implicated TRAF6 as important for thymic Treg generation and *Foxp3* expression in peripheral lymphoid tissues (Shimo et al, 2011). More recently, unstable expression of FOXP3 and defective *in vivo* Treg function have been reported in TRAF6-deficient Tregs (Muto et al, 2013). These results suggest that TRAF6 expression in the Treg compartment is necessary for an optimal pool of these suppressor cells.

We next explored the effects of TRAF6 knockout during Treg differentiation. Prior work has shown that T-cell-restricted TRAF6 deficiency does not impair *in vitro*, TGFβ-driven induction of FOXP3 expression (Cejas et al, 2010; Muto et al, 2013). Also, despite displaying stunted thymic Treg development, mice globally deficient in TRAF6 yield T cells that are actually more inclined to FOXP3 induction *in vitro* (Shimo et al, 2011). We found that activation of naïve $CD4^+$ T cells from wild-type and $Traf6^{fl/fl}Foxp3Cre^+$ mice in the presence of TGFβ and IL-2 resulted in similar levels in *in vitro* FOXP3 induction, even when suboptimal concentrations of TGFβ were used (Fig 1G and J). However, in other experiments, addition of the proinflammatory, Th17-inducing cytokine, IL-6, disrupted iTreg skewing even under potent Treg-inducing conditions (i.e., 5 ng/ml TGFβ, 100 U IL-2/ml). This effect was evidenced by low levels of FOXP3 induction in the IL-6-exposed cells. This was seen to an even greater degree when TRAF6 was deleted in the newly induced Tregs (Fig EV1A and B). Interestingly, when fully differentiated iTregs, which can be prone to unstable *Foxp3* expression, were treated with proinflammatory cytokines, FOXP3 protein levels decreased more readily in the absence of TRAF6 expression. This was particularly the case for IL-6 exposure, while TNF-α and IL-1β treatment reduced FOXP3 expression in both groups (Fig EV1C and D). Treatment with the TLR ligand LPS also disproportionately reduced FOXP3 levels in $Traf6^{fl/fl}Foxp3Cre^+$-derived iTregs (Fig EV1E). These findings suggest that TRAF6 expression in Tregs may work to stabilize or bolster expression of FOXP3 in multiple Treg subsets in the face of inflammatory stresses.

### Expression of TRAF6 is required for *in vivo* Treg suppression

We next investigated the role of TRAF6 activity on the suppressive function of Tregs. To this end, we isolated Tregs from wild-type ($Traf6^{fl/fl}Foxp3Cre^-$) and $Traf6^{fl/fl}Foxp3Cre^+$ mice by FACS and compared their suppressive potency using an *in vitro* suppression assay. The ability of these Tregs to suppress the proliferation of naïve $CD4^+$ "responder" T cells was determined by measuring CFSE dilution by flow cytometry. We found that TRAF6-deficient Tregs were functionally comparable to wild-type Tregs *in vitro* (Appendix Fig S2A and B) in agreement with prior studies of Tregs isolated from $Traf6^{fl/fl}CD4Cre^+$ mice (King et al, 2006). In contrast, an *in vivo* assay of Treg function revealed that without TRAF6, Tregs failed to restrain responder T-cell expansion. Here, $CD45.2^+$ Tregs from either $Traf6^{fl/fl}Foxp3Cre^+$ or wild-type mice were purified and mixed with normal naïve responder $CD4^+$ T cells ($CD45.1^+$) at a 1:5 ratio prior to injection into lymphopenic $Rag2^{-/-}$

---

**Figure 1.    TRAF6 is highly expressed by Treg subsets and plays an important role in maintaining immune homeostasis.**

A    TRAF6 expression in differentiating $CD4^+$ T cells. Naïve $CD4^+$ T cells were obtained from wild-type C57BL/6 mice by FACS and activated with anti-CD3/CD28 (1 μg and 2 μg/ml) for the indicated times in the presence of distinct T helper lineage-directing cytokines or under neutral activation conditions (Th0). After total RNA extraction and cDNA conversion, RT–PCR determined Th0, Th1, Th17, and iTregs.

B    *TRAF6* mRNA expression by human Tregs and non-Treg $CD4^+$ T cell. Human Tregs ($CD3^+/CD4^+/CD8^-/CD25^{HIGH}/CD127^{low}/CD39^+$) and non-Treg $CD4^+$ T cells ($CD3^+/CD4^+/CD8^-/CD25^-$) were obtained from the peripheral blood of healthy donors by FACS after Ficoll–Paque PLUS gradient centrifugation and magnetic bead enrichment of $CD4^+$ T cells. *TRAF6* mRNA was measured by qRT–PCR.

C, D    Evidence of lymphoproliferative disease in $Traf6^{fl/fl}Foxp3Cre^+$ mice. (C) Spleens and lymph nodes were recovered from $Traf6^{fl/fl}Foxp3Cre^+$ mice and $Traf6^{fl/fl}$ littermates at 8 weeks of age (Scale bars: 5 mm). (D) The cellularity of the lymphoid tissues of $Traf6^{fl/fl}Foxp3Cre^+$ mice and their $Traf6^{fl/fl}$ $Foxp3Cre^-$ littermates was determined (eight mice/group).

E, H    Effect of Treg-specific TRAF6 deficiency on baseline T-cell activation. The frequencies of effector cells ($CD44^{high}/CD62L^{low}$), memory cells ($CD44^{high}/CD62L^{high}$), and naïve cells ($CD44^{low}/CD62L^{high}$) in the $CD4^+$ T-cell compartments of $Traf6^{fl/fl}$ and $Traf6^{fl/fl}Foxp3Cre^+$ mice were determined by flow cytometry (five mice/group).

F, I    Effect of Treg-specific TRAF6 deficiency on baseline T-cell activation. The frequencies of effector cells ($CD44^{high}/CD62L^{low}$), memory cells ($CD44^{high}/CD62L^{high}$), and naïve cells ($CD44^{low}/CD62L^{high}$) in the $CD8^+$ T-cell compartments of $Traf6^{fl/fl}$ $Foxp3Cre^-$ (wild type) and $Traf6^{fl/fl}Foxp3Cre^+$ mice were determined by flow cytometry (five mice/group).

G, J    Impact of TRAF6 expression on *in vitro* Treg differentiation. As in (A), naïve $CD4^+$ T cells were isolated from $Traf6^{fl/fl}Foxp3Cre^+$ and $Traf6^{fl/fl}$ mice and differentiated into iTregs. Conditions of suboptimal TGFβ concentrations (0.5, 0.05 ng/ml) were tested as well, and intracellular FOXP3 was measured after 4 days.

Data information: Panels (A, B, D, H, I, and J) represent mean results ± SEM. *$P < 0.05$; **$P < 0.01$; ***$P < 0.001$; ****$P < 0.0001$; ns, no significance, by an unpaired *t*-test. (A, C, E–G) are representative findings from at least three experiments. Panels (A, J) represent the means of three replicates.

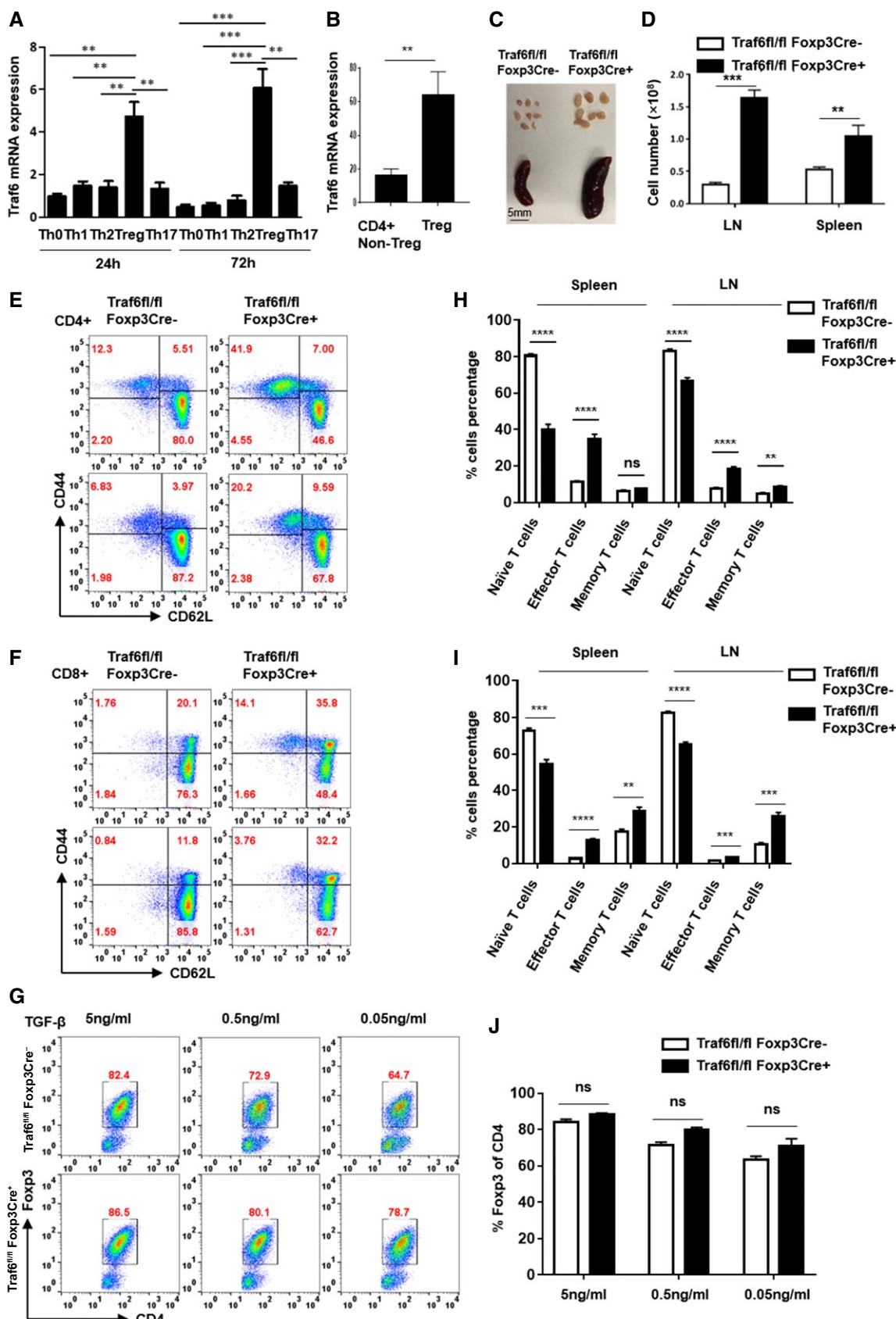

**Figure 1.**

recipient mice. After 7 days, the relative frequencies and numbers of both responder and Treg cells in the spleens of recipient mice were observed by flow cytometry. Responder cells injected with Traf6$^{fl/fl}$Foxp3Cre$^+$-derived Tregs were much more plentiful than those co-transferred with wild-type Tregs, and knockout Tregs were scarcer than their wild-type counterparts (Fig 2A). These results suggest that TRAF6 is necessary for Treg suppressive function *in vivo*, also in agreement with the findings of prior studies (Shimo *et al*, 2011; Muto *et al*, 2013). Interestingly, the dysfunction seen in the absence of TRAF6 was not accompanied by disrupted expression of Treg-associated factors. For instance, levels of GITR and CTLA-4 were somewhat elevated in the FOXP3$^+$ cells of Traf6$^{fl/fl}$Foxp3Cre$^+$ mice relative to wild-type controls. This was likely indicative of a hyperactivated T-cell compartment rather than a Treg population with heightened function. Staining patterns of CD44 and CD62L on Traf6$^{fl/fl}$Foxp3Cre$^+$-derived cells were consistent with this notion (Appendix Fig S2C). These findings reinforce the link between TRAF6 expression by Tregs and suppressive function.

## TRAF6 deficiency enhances anti-tumor immunity and slows the progression of B16 melanomas

We next explored how TRAF6 modulation could affect Treg-enforced immune suppression in the cancer setting. To this end, we challenged Traf6$^{fl/fl}$Foxp3Cre$^+$ mice and their TRAF6-competent littermates with aggressive, poorly immunogenic B16 melanomas. After subcutaneous (s.c.) injection, we measured the growth of the implanted tumors. Confirming the inability of TRAF6-deficient Tregs to enforce tumor-associated immune tolerance, Traf6$^{fl/fl}$Foxp3Cre$^+$ mice failed to support the growth of implanted B16 melanoma cells. The tumors of wild-type mice, on the other hand, readily developed and grew progressively (Fig 2B). In keeping with the notion that TRAF6 is necessary for Treg-mediated immune restrain, the severely stunted tumor progression in Traf6$^{fl/fl}$Foxp3Cre$^+$ mice coincided with a markedly enhanced anti-tumor response. This was evidenced by a heightened production of the proinflammatory cytokines IFN-γ and IL-17 by leukocytes infiltrating the tumor and tumor-draining lymph node (Fig 2C and D). Additionally, deficiencies in Treg TRAF6 expression lead to reduced FOXP3$^+$ T-cell frequencies within the CD4$^+$ T cells infiltrating several lymphoid and tumor tissues (Fig 2E and F). Similar results were obtained in the more immunogenic MC38 colon cancer model. Specifically, we found that

Treg-restricted TRAF6 deficiency severely delayed implanted tumor growth (Figs EV2A and 2B) while enhancing anti-tumor immunity (i.e., boosting frequencies of proinflammatory cytokine-producing T cells) at the expense of Treg presence in the tumor microenvironment (Fig EV2C). These findings clearly illustrate the important role played by TRAF6 in Treg-mediated immune control and tumor-enforced suppression of anti-cancer immunity. These findings suggest that potential therapies targeting TRAF6 may be highly effective at breaking tolerance and bolstering the anti-tumor immunity by undermining Treg function.

## TRAF6 interacts with FOXP3 and mediates K63 polyubiquitination of the transcription factor

Posttranslational modifications can impact the expression and function of FOXP3 in Tregs, and several examples recently have been described (van Loosdregt & Coffer, 2014; Li *et al*, 2015). Previously, we found that FOXP3 was subject to K48-linked polyubiquitination at lysines 227, 250, 263, and 268, resulting in FOXP3 degradation (Chen *et al*, 2013). While the tagging of target proteins with K48-linked polyubiquitin chains by E3 ligases generally leads to degradation, modification of proteins with chains of ubiquitins interlinked at other lysine residues (K63 for instance) can yield non-proteolytic outcomes. Such modifications are typically associated with altered intracellular trafficking or changes in the activity of target proteins. This is indeed known to be the case for a number of factors with demonstrated importance to leukocyte function and immune activation (Chen & Sun, 2009). Prior to our finding that TRAF6 is apparently capable of promoting FOXP3 ubiquitination, the enzyme was widely known as an important facilitator of K63 ubiquitination (Jiang & Chen, 2011; Geng *et al*, 2015; Hu & Sun, 2016). Potential K63-type modifications of FOXP3 and their consequences for the fate of the transcription factor and Treg function have yet to be studied. We therefore set out to determine whether TRAF6, a well-characterized facilitator of K63 ubiquitination, executes its significant contribution to Treg phenotype and function through a hitherto unappreciated, regulatory mechanism acting on FOXP3 protein.

To this end, we co-expressed TRAF6 and FOXP3 in cell lines that also expressed a particular variant of FLAG-labeled ubiquitin—one in which all lysine residues had been lost (mutated to ubiquitin-resistant arginine residues) save for K63 ("FLAG-Ub-K63"). Indeed, robust ubiquitination of FOXP3 was observed only when TRAF6

**Figure 2. Expression of TRAF6 by Tregs is required for *in vivo* function and suppression of anti-tumor immunity.**

A    *In vivo* suppressive function of WT and Traf6$^{fl/fl}$Foxp3Cre$^+$ Tregs. Tregs from the indicated mice (CD45.2$^+$) were isolated by FACS, as were naïve (CD62L$^{high}$/CD25$^-$) CD4$^+$ responder T cells from congenically distinct (CD45.1$^+$) donor mice. Tregs and Tresponders were mixed at a 1:5 (2 × 10$^5$:10 × 10$^5$) ratio before injection into Rag2$^{-/-}$ mice. 7 days later, spleens were harvested, and the relative frequencies and absolute numbers of each transferred cell populations were determined by flow cytometry.

B    Implanted B16 melanoma growth in Traf6$^{fl/fl}$Foxp3Cre$^+$ and wild-type (Traf6$^{fl/fl}$Foxp3Cre$^-$) mice. 1 × 10$^5$ B16F10 cells were injected subcutaneously (s.c.) into the shaved flanks of the indicated mice (*n* = 5/group). Tumor volumes were monitored every 3 days.

C, D    Proinflammatory cytokine production in tumor-bearing mice with and without Treg-specific TRAF6 expression. Cell suspensions of the tumor-draining lymph node and tumor-infiltrating leukocytes (TILs) were recovered after 21 days of tumor growth. *Ex vivo* stimulation with PMA and ionomycin in the presence of Golgistop for 5 h preceded intracellular staining for IFN-γ and IL-17 and flow cytometry analysis.

E, F    FOXP3 expression by CD4$^+$ T cells in tumor-bearing Traf6$^{fl/fl}$Foxp3Cre$^+$ and wild-type (Traf6$^{fl/fl}$Foxp3Cre$^-$) mice. The frequency of FOXP3$^+$ cells in the CD4$^+$ cells of tumor-draining lymph nodes (dLN), peripheral, non-tumor-draining lymph nodes (pLN), spleens, and TILs was determined by flow cytometry.

Data information: (A right, B, D, E, and F) denote the average results from at least three independent experiments ± SEM (3–5 replicates).*P < 0.05; **P < 0.01; ***P < 0.001 by an unpaired *t*-test. Panels (A left, C and E) show representative findings from these experiments.

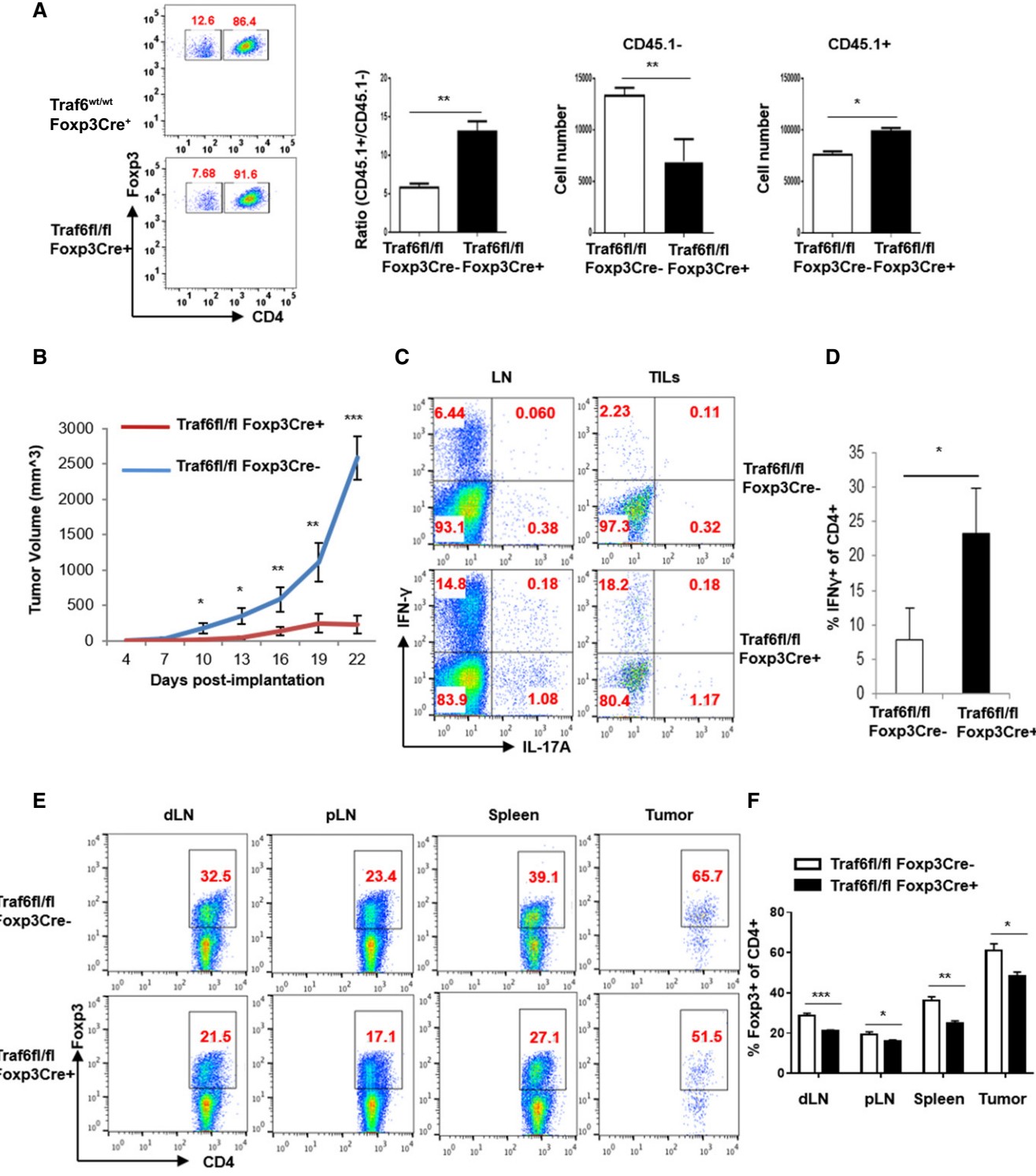

**Figure 2.**

was co-expressed in these cells. Furthermore, nullifying the E3 ligase activity of TRAF6 through mutation of its RING domain (C70A) mostly ablated the ability of this enzyme to induce K63-type ubiquitination of FOXP3 in cell lines (Fig 3A). Another previously reported TRAF6 mutant with a more specific defect in its E3 ligase activity (and lacking the more extensive structural changes and ligase-independent effects of the C70A point mutation (Strickson et al, 2017)) proved similarly unable to execute this particular modification of FOXP3 protein (Fig 3A). These findings implicate TRAF6 as an E3 ligase responsible for K63-linked ubiquitination of FOXP3.

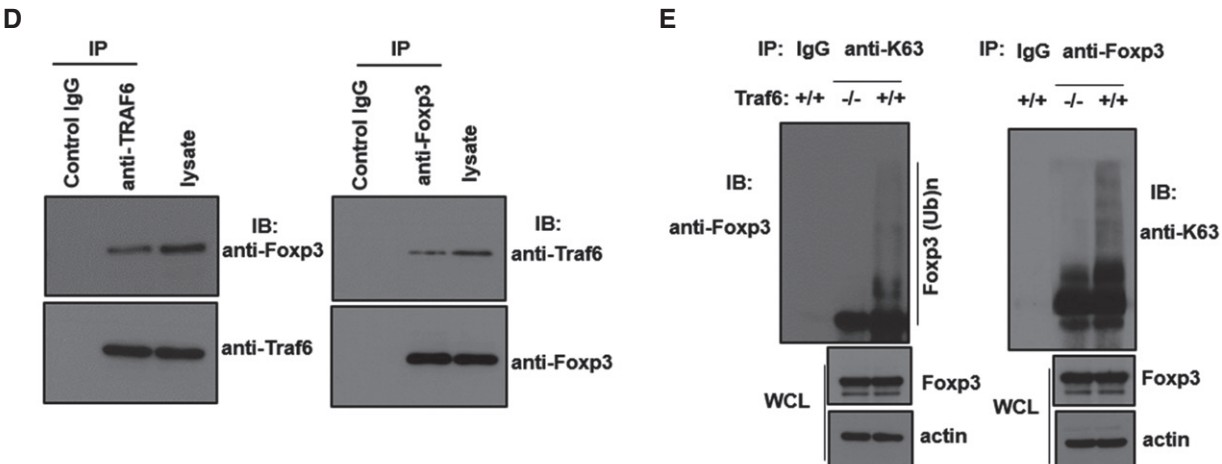

**Figure 3.  TRAF6 physically interacts with FOXP3 and induces K63-linked polyubiquitination of the transcription factor.**

A  Assessment of K63-type ubiquitination of FOXP3 upon expression of wild-type and catalytically deficient TRAF6 mutants. 293T cells were transfected with the given combinations of vectors encoding wild-type TRAF6, the enzymatically deficient L74H or C70A TRAF6 mutants, HA-FOXP3, and FLAG-labeled ubiquitin molecules possessing a single lysine residue (K63) that restrain the possible ubiquitin monomer linkages on polyubiquitinated proteins to the K63-type only. Cells were lysed, and proteins modified by K63-ubiquitin chains were recovered from lysates by pulling down using a bead-immobilized K63-specific TUBE reagent. The presence of FOXP3 among these modified proteins was determined by immunoblotting as were levels of FOXP3 and TRAF6 in the pre-IP whole cell lysate (WCL).

B  Co-immunoprecipitation (co-IP) of FOXP3 with TRAF6 and its facilitation of K63-linked ubiquitination. 293T cells expressing the indicated constructs encoding HA-FOXP3 and FLAG-TRAF6 were lysed and incubated with bead-immobilized anti-FLAG antibodies. FOXP3 molecules co-IPed in this manner were resolved by SDS–PAGE and detected by immunoblotting with anti-HA antibodies.

C  293T cells were transfected with the indicated combinations of expression constructs encoding HA-FOXP3, Myc-TRAF6, and FLAG-tagged ubiquitin molecules—either wild-type (FLAG-Ub) or a variant with a K-to-R mutation at either lysine residue 48 (K48R) or 63 (K63R) responsible for preventing K48- and K63-type polyubiquitination, respectively.

D  Endogenous, reciprocal co-IP of FOXP3 and TRAF6 in murine iTreg lysate. Naïve CD62L$^{high}$/CD25$^-$/CD4$^+$ T cells were obtained by FACS and activated with anti-CD3/CD28 antibodies (1 μg and 2 μg/ml, respectively) in the presence of IL-2 and TGFβ (100 U/ml and 5 ng/ml, respectively) for 4 days before lysis. Antibodies against TRAF6 (left) or FOXP3 (right) were used to pull down target proteins and their interaction partners, which were resolved under denaturing conditions and visualized by immunoblotting.

E  Degree of K63 ubiquitination in the cellular FOXP3 pools of Traf6$^{fl/fl}$Foxp3Cre$^+$ and Traf6$^{wt/wt}$Foxp3Cre$^+$ mice. CD4$^+$/YFP$^+$ cells from Traf6$^{fl/fl}$Foxp3Cre$^+$ ("Traf6$^{-/-}$") or wild-type Traf6$^{wt/wt}$Foxp3Cre$^+$ mice ("Traf6$^{+/+}$") were isolated by FACS and lysed, and either total FOXP3 (right) or K63-ubiquitin-modified proteins (left) were immunoprecipitated. Levels of polyubiquitinated FOXP3 were observed in each by probing with antibodies specific for anti-K63 ubiquitin and FOXP3, respectively, and levels of total FOXP3 and actin were also measured by immunoblot analysis of whole cell lysate.

Data information: Shown are the representative findings of three independent experiments. Where indicated, levels of FOXP3, TRAF6, or actin (loading control) were measured in the pre-IP WCL input as well.
Source data are available online for this figure.

To further explore the potential ligase–target relationship between TRAF6 and FOXP3, we tested whether these factors physically interact. Here, we performed co-immunoprecipitation (co-IP) experiments with the lysates of the 293T cell lines transfected with expression vectors encoding HA-tagged FOXP3 and FLAG-TRAF6. Upon immunoprecipitation of TRAF6, we detected associated FOXP3 protein among the material recovered from lysate (Fig 3B), suggesting that TRAF6 indeed interacts with this key Treg transcription factor.

Co-expression of HA-FOXP3 and Myc-TRAF6 constructs along with those encoding distinct FLAG-tagged ubiquitin variants allowed us to confirm the nature of the FOXP3 modification arising from this apparent interaction. Pull-down of labeled FOXP3 from cell lysates by anti-HA antibody facilitated examination of the various ubiquitinated forms of the transcription factor by immunoblot analysis. Considerable ubiquitination of FOXP3 was seen in cells triple-transfected with expression constructs encoding FOXP3, TRAF6, and wild-type ubiquitin (Fig 3C). Those lacking ectopic TRAF6 expression, however, showed only minor levels of FOXP3 polyubiquitination. Interestingly, expression of ubiquitin molecules possessing mutated lysine residue K48 (K48R), which are incapable of supporting K48 polyubiquitination, still resulted in robust FOXP3 ubiquitination in this system. Constructs encoding ubiquitin monomers having a lysine-to-arginine (K-to-R) mutation at residue K63 (K63R), on the other hand, led to a marked reduction in FOXP3 ubiquitination (Fig 3C). These findings suggest that TRAF6 facilitates a previously unstudied K63-type ubiquitination of FOXP3 likely relevant in the biology of Tregs. This notion was confirmed in primary murine iTregs using a reciprocal endogenous co-IP approach. In these experiments, we found that TRAF6 was capable of pulling down associated FOXP3 protein from iTreg lysates and vice versa (Fig 3D). We also assessed the extent of K63 ubiquitination present within the FOXP3 protein pools of freshly isolated murine nTreg. Polyubiquitinated FOXP3 species were readily detected among the K63-ubiquitin-modified proteins pulled down from Treg lysates by anti-K63 ubiquitin antibodies. Likewise, this modification was also seen following pull-down with anti-FOXP3 antibodies (Fig 3E). Importantly, we also found that the dysfunctional TRAF6-deficient Tregs of Traf6$^{fl/fl}$Foxp3Cre$^{+}$ mice (Traf6$^{-/-}$) displayed a relative dearth of K63-ubiquitinated FOXP3 species compared to normal Tregs (Traf6$^{+/+}$; Fig 3E), implicating that this particular modification of FOXP3 by TRAF6 may be critical for optimal Treg function.

### The zinc finger and leucine zipper domains of FOXP3 and lysine residue 262 are necessary for interaction with TRAF6 and ubiquitination

We then set out to further characterize the interaction between FOXP3 and TRAF6. To identify the TRAF6-interacting domain of FOXP3, we generated deletion mutants each lacking one or several of the recognized functional domains of the transcription factor (Fig EV3A–C). The ability of full-length FOXP3 and each deletion variant to interact with TRAF6 was determined by co-IP approaches. Molecules that contained intact zinc finger and leucine zipper domains (full-length Foxp3, N1, N2, and C3) readily pulled down TRAF6. Meanwhile, those lacking these domains (N3, C1, and C2) failed to do so (Fig EV3D). These results reveal that the zinc finger

and leucine zipper domain of FOXP3 are indispensable for interaction with TRAF6. We next assessed the ability of these deletion mutants to become modified by K63 ubiquitination. To this end, we lysed cells co-expressing TRAF6 and the various aforementioned *Foxp3* constructs and visualized the extent of K63 ubiquitination present among the full-length or mutant FOXP3-HA protein pools by immunoblot analysis. As expected, only FOXP3 species capable of TRAF6 interaction (and possessing intact zinc finger/leucine zipper domains) displayed K63 ubiquitination. Curiously, a C-terminal deletion mutation retaining these regions but lacking the proline-rich domain ("C3") also displayed poor K63 ubiquitin signal (Fig EV3E). This could reflect the proximity of the target residue to the proline-rich domain or a conformational issue preventing modification of the truncated protein.

In order to identify potential target sites in the FOXP3 molecule subject to TRAF6-mediated ubiquitination, we also screened *Foxp3* constructs with various K-to-R mutations for resistance to modification by this enzyme. As before, prevalent ubiquitination was seen in the FOXP3 pool in the presence of TRAF6. As expected, we also found that mutating all twenty lysine residues (20R) largely prevented the ubiquitination of FOXP3 by TRAF6. Individual constructs encoding unique single lysine residue-containing FOXP3 variants (e.g., K8, K31, K144), could, to varying degrees, support FOXP3 ubiquitination in the presence of TRAF6 (Fig 4A). Of these single-lysine mutants, however, K262 alone was able to fully support TRAF6-mediated ubiquitination of the FOXP3 target protein. In contrast, mutation of Foxp3 at K262 (K262R) ablates TRAF6-mediated ubiquitination of FOXP3 (Fig 4B). These results shed light on the nature of the TRAF6-FOXP3 interaction, and in particular identify the specific residue on the FOXP3 molecule modified as a result of this pairing.

### TRAF6 target residue K262 is required for nuclear FOXP3 localization and regulatory function

We then assessed how the loss of K63 ubiquitination affected the regulatory function of FOXP3. A gene reporter assay revealed, expectedly, that sequestering FOXP3 in the perinuclear region severely undermined the ability of the transcription factor to regulate gene expression. IL-2 expression is known to be among the genes directly suppressed by FOXP3 in Tregs (Lopes *et al*, 2006; Zheng *et al*, 2007). We therefore compared the ability of normal FOXP3 and the K262R mutant to silence expression of a luciferase reporter gene under the control of the *Il2* gene promoter. While a normal *Foxp3* expression vector significantly blocked the expression of the luciferase reporter, the K63-ubiquitin-resistant K262R mutant failed to do so—resembling the lack of suppression seen in an empty vector control (Fig 5A). This indicated that without K63 modification, FOXP3 is functionally impaired in its capacity to repress effector T-cell gene expression in Tregs.

We confirmed these results in primary murine CD4$^{+}$ T cells by ectopically expressing either a normal Foxp3 construct or one encoding the K262R mutant (delivered by retroviral transduction). Here, recipient cells were activated, and we measured IL-2 production by ELISA. We found that while normal FOXP3 expression could effectively repress IL-2 expression by wild-type naïve CD4$^{+}$ T cells, K262R expression could not. Importantly, when TRAF6 was absent in these T cells (i.e., when they were obtained from Traf6$^{fl/fl}$

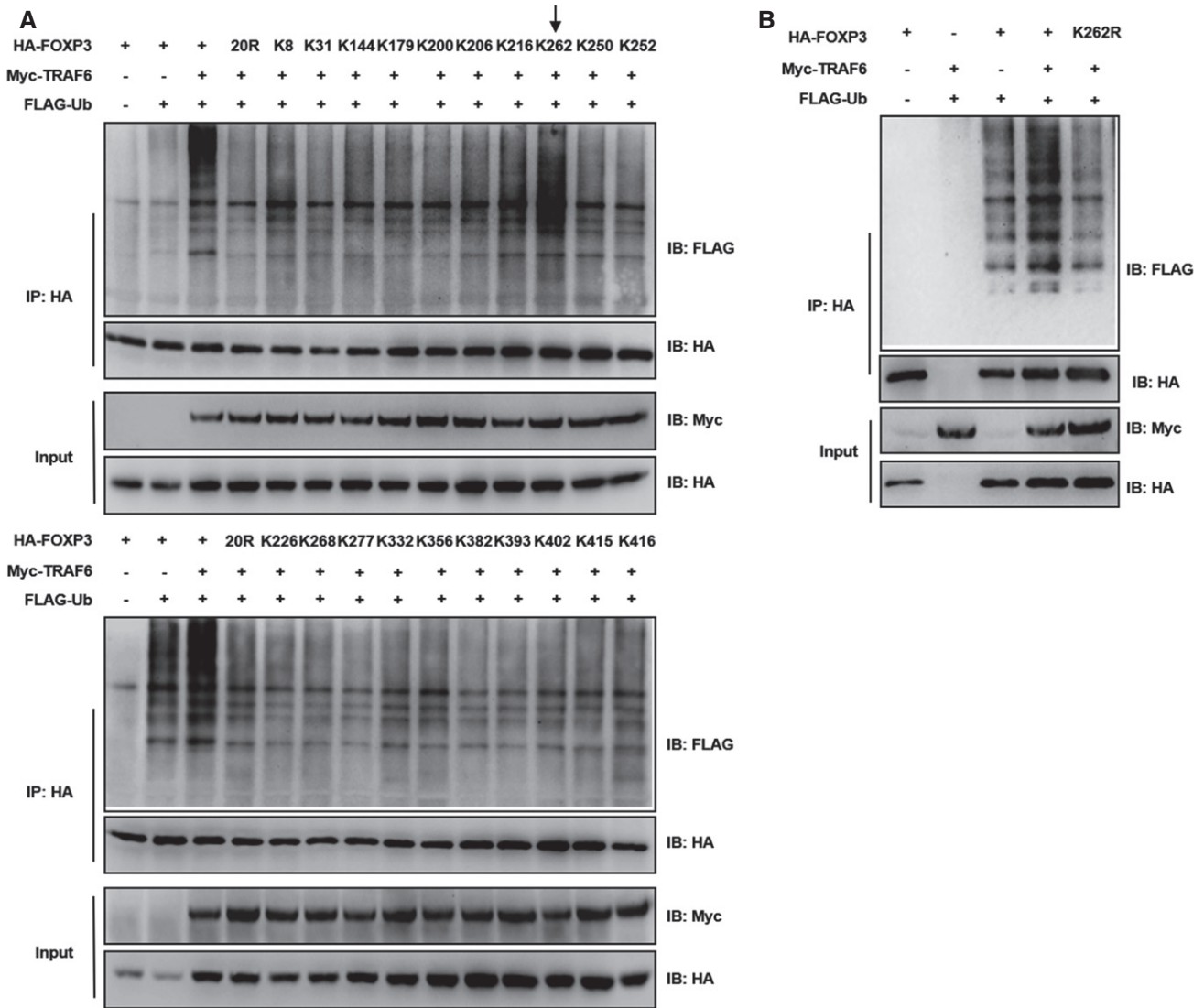

**Figure 4. TRAF6 targets lysine residue 262 on FOXP3 for ubiquitination.**

A   Immunoblot analysis of TRAF6-mediated ubiquitination of wild-type and mutant FOXP3 molecules. 293T cell lines were transfected with a normal HA-tagged *Foxp3* construct, another encoding a ubiquitination-resistant mutant in which all lysine residues were replaced by arginines (20R), or one of 20 single lysine-containing constructs with only the indicated lysine residue available for modification. These cell lines also received expression vectors encoding TRAF6 and ubiquitin molecules labeled with Myc and FLAG tags, respectively. Negative controls did not receive TRAF6 or expressed FOXP3 alone. Labeled FOXP3 proteins were pulled down from cell lysates (anti-HA), and ubiquitinated species were visualized by immunoblotting for FLAG.

B   Assessing the impact of K262 mutation on FOXP3 ubiquitination by TRAF6. As in (A) 293T cells were transfected with combinations of expression constructs encoding Myc-TRAF6, FLAG-ubiquitin, and an HA-tagged FOXP3 molecule that was either normal (HA-FOXP3) or possessed a K-to-R mutation at residue 262 (K262R). FOXP3 proteins were pulled down with anti-HA beads, and either ubiquitinated proteins or total FOXP3 proteins were detected by probing for FLAG and HA, respectively.

Data information: Shown are representative blots from three experiments. Expression of FOXP3 and TRAF6 by transfectants pre-IP WCL was confirmed (input).
Source data are available online for this figure.

CD4Cre$^+$ donors), even delivery of wild-type *Foxp3* expression vector could not suppress IL-2 production (Fig 5B). This nicely illustrates the importance of both TRAF6 and its target residue for proper control of Treg gene transcription by FOXP3.

Since K63-type ubiquitination can dictate the intracellular distribution of target proteins and TRAF6 was previously implicated in promoting K63-type ubiquitination and nuclear localization of another protein, NRIF (Geetha *et al*, 2005), we hypothesized that

modification of FOXP3 by TRAF6 in this manner also promotes the nuclear localization of FOXP3. To test this, we generated HeLa cells expressing the K262R FOXP3 mutant that was resistant to K63 ubiquitination even in the presence of TRAF6. Immunofluorescence (IF) microscopy suggested that the K262R mutant was not able to localize to the nucleus in contrast to wild-type FOXP3 protein (Fig 5C). These results suggest that TRAF6-facilitated modification at FOXP3's K262 site is a molecular event central to the proper intracellular

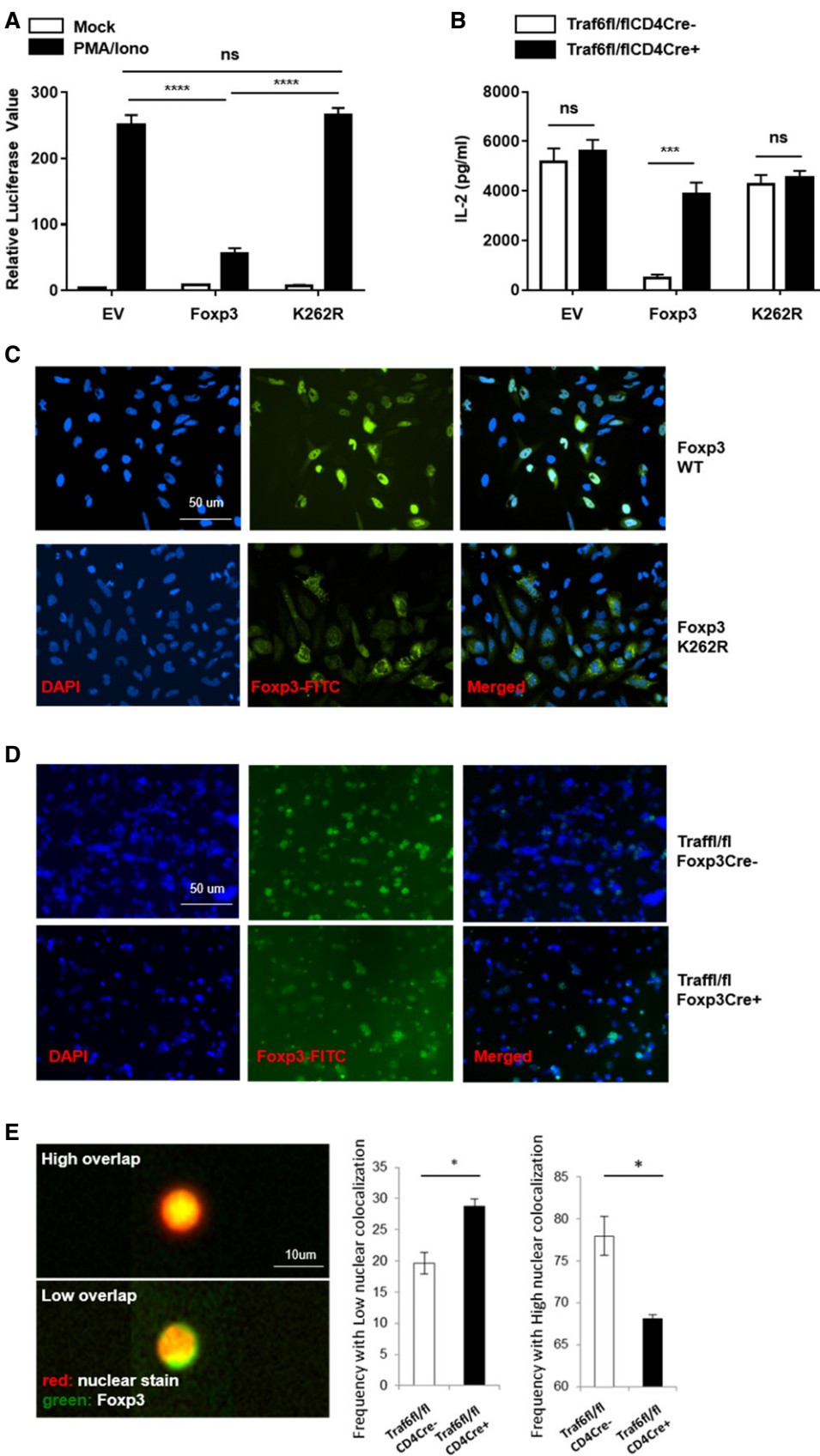

**Figure 5.**

**Figure 5.  Ablation of TRAF6 activity or the K63-type ubiquitination site in Foxp3 disrupts its cellular distribution, expression, and regulatory function.**

A   Impact of K63 ubiquitination loss on FOXP3's gene silencing capacity. Jurkat T cells transfected with a wild-type *Foxp3* expression vector, one encoding the K262R mutant, or an empty (control) vector also received a dual-luciferase reporter construct where luciferase expression was under the control of the *Il2* promoter. After activation with PMA and ionomycin for 8 h, luciferase activity was assayed.

B   IL-2 production in the presence of wild-type and K63-ubiquitination-resistant FOXP3. Naïve CD4$^+$ T cells isolated from Traf6$^{fl/fl}$CD4Cre$^+$ or WT mice were activated overnight by anti-CD3/CD28 antibodies, then transduced by retroviruses carrying normal *Foxp3*, the K262R mutant, or an empty vector. Ires-GFP in the retroviral vector serves as an internal control. GFP$^+$ cells were sorted out 48 h posttransduction. The sorted cells were cultured for one additional day. Culture supernatants were collected for measuring the levels of IL-2 by ELISA.

C   Cellular distribution of wild type and K262R FOXP3. HeLa cells were transfected with expression vectors encoding either wild-type FOXP3 or the K262R mutant. The relative overlap of FOXP3 protein signal and DAPI-stained nuclei was observed by fluorescence microscopy (blue: DAPI, green: FOXP3, Scale bars: 50 μm).

D   Immunostaining of FOXP3 in Traf6$^{fl/fl}$CD4Cre$^+$ and wild-type (Traf6$^{fl/fl}$CD4Cre$^-$)-derived suspensions of lymph node and spleen cells (blue: DAPI, green: FOXP3, Scale bars: 50 μm).

E   Quantification of FOXP3 distribution in murine Tregs. The degree of FOXP3-nuclear colocalization in CD4$^+$/FOXP3$^+$ cells isolated from the lymph nodes and spleens of Traf6$^{fl/fl}$CD4Cre$^+$ and wild-type (Traf6$^{fl/fl}$CD4Cre$^-$) mice was determined by ImageStream analysis, and the frequencies of cells with either a low or high probability of nuclear FOXP3 distribution are shown (red: nuclear stain, green: FOXP3, Scale bars: 10 μm).

Data information: (A, B, and E right) represent mean results from three experiments ± SEM. *$P < 0.05$; **$P < 0.01$; ***$P < 0.001$; ****$P < 0.0001$; ns, no significance (unpaired *t*-test). (C, D) show representative staining results, while (E left) depicts exemplars for high and low nuclear overlap from ImageStream analysis.

localization of this important transcription factor. Interestingly, when we stained HeLa cells containing both FOXP3-K262R- and TRAF6-encoding constructs, we observed overlapping signal suggesting that the mutant FOXP3 retained the ability to colocalize with TRAF6, albeit in the perinuclear region (Fig EV4A). Indeed, reciprocal co-IP experiments confirmed that, while resistant to K63 ubiquitination by TRAF6, the K262R mutant FOXP3 protein is still able to physically interact with the E3 ligase (Fig EV4B). These findings suggest that the interaction of FOXP3-TRAF6 and the enzymatic modification of FOXP3 are indeed distinct molecular events.

To further test the notion that TRAF6's E3 ligase function was central to both the K63 ubiquitination of FOXP3 and its activity, we ectopically expressed wild-type TRAF6 and the enzymatically inactive mutants L74H and C70A in genetically TRAF6-deficient CD4$^+$ T cells that also received a *Foxp3* expression vector. These cells were readily found to contain K63-ubiquitinated FOXP3 species when reconstituted with wild-type TRAF6. However, the ligase-inactive TRAF6 mutant was unable to rescue defects in this form of modification (Fig EV4C). Introduction of ectopic *Foxp3* into Traf6$^{fl/fl}$CD4Cre$^+$-derived CD4$^+$ T cells also failed to optimally restrain IL-2 production by cells also transduced with mutant TRAF6, unlike control cells with truly reconstituted TRAF6 activity (Fig EV4D). With the phenotypes of K63 ubiquitination resistance (K262R) largely recapitulated by TRAF6 ligase mutants, the evidence in favor of this enzyme's role as a previously unappreciated modifier of FOXP3 protein was thus bolstered.

Importantly, we also observed perturbations of FOXP3's intracellular distribution in the absence of TRAF6 activity in primary murine T cells. Here, Tregs isolated from Traf6$^{fl/fl}$Foxp3Cre$^+$ and Traf6$^{fl/fl}$CD4Cre$^+$ mice displayed aberrant, perinuclear accumulation of FOXP3 while their respective normal littermates did not (Fig 5D and E). Additionally, the dysfunction of TRAF6-deficient Tregs in our tumor experiments was found to be associated with this uncharacteristic distribution of FOXP3 protein. Visualization of intracellular FOXP3 among the splenocytes and tumor-infiltrating leukocytes from tumor-bearing TRAF6$^{fl/fl}$Foxp3Cre$^+$ mice revealed that most of the factor detected was found in extranuclear deposits, in stark contrast to the nuclear staining expected and seen in wild-type controls (Fig EV5A). Confocal microscopy of iTregs generated

from the naïve precursors of wild-type (Traf6$^{fl/fl}$Foxp3Cre$^-$) and Traf6$^{fl/fl}$Foxp3Cre$^+$ mice confirmed that in the absence of TRAF6-mediated FOXP3 modification, normal distribution of this key transcription factor is markedly disrupted (Fig EV5B).

To shed light on the potential fate of extranuclear FOXP3 protein deprived of K63 modification, we assessed the stability of both normal and K262R FOXP3 protein pools *in vitro*. Here, FOXP3 protein levels were assessed in cell lines carrying wild-type and mutant *Foxp3* constructs that were treated with cyclohexamide (CHX) over time, while K262R-FOXP3 levels were depleted much more rapidly compared to wild-type FOXP3 (Appendix Fig S3A). In line with these findings, TRAF6 deficiency in primary Tregs similarly enhanced the FOXP3 turnover rate beyond that seen in wild-type controls (Appendix Fig S3B). These findings are in harmony with those previously reported regarding the instability of FOXP3 expression in the absence of TRAF6 (Muto *et al*, 2013). They also reveal a potential connection between the processes of FOXP3 localization, K63 ubiquitination, and protein level control over the expression of this transcription factor.

These results suggest that potential therapies capable of targeting TRAF6-mediated K63 ubiquitination in Tregs may be highly effective at breaking tolerance and bolstering the anti-tumor immunity by undermining Treg function. Additional exploration of the relative importance of this newly uncovered role for TRAF6 and its other potential contributions to the tumor–immune cell interface, however, will be needed.

### Tregs insensitive to K262 ubiquitination are dysfunctional *in vitro* and *in vivo*

These findings strongly suggest that ubiquitination of FOXP3 at K262 is key for this factor's ability to anchor the gene expression patterns responsible for Treg function. In line with both compromised FOXP3 function and instability in the absence of K63 ubiquitination, we found that cells expressing the K262R *Foxp3* mutant were broadly dysfunctional in assays of suppressive function. *In vitro*, CD4$^+$ T cells transduced with lentiviral *Foxp3* expression vector effectively dampened the proliferation of co-cultured naïve responder cells. In contrast, *Foxp3*-K262R-expressing cells were

less effective suppressor cells (Fig 6A and B). We also explored the impact of K63 resistance on Treg function *in vivo* using a T-cell-dependent mouse model of colitis. Here, adoptive transfer of naïve CD4$^+$ T cells (Thy1.2/CD90.2$^+$) into Rag2$^{-/-}$ mice induced gut inflammation and progressive wasting disease in recipients. Co-injection of normal Tregs along with the colitogenic T cells effectively prevented the development of disease. CD4$^+$ T cells ectopically expressing wild-type *Foxp3* were also largely protective in this model. *Foxp3*-K262R transductants, on the other hand, failed to prevent severe colitis and mice receiving these "engineered Tregs" resembled no-Treg controls in terms of weight loss (Fig 6C). These trends were also seen in the degree of large bowel histopathology observed across these groups (Fig 6D and E). In line with these observations, mice that received no Tregs harbored considerable numbers of expanding T effector cells infiltrating their spleens, mesenteric lymph nodes, and lamina propria. Meanwhile, the numbers and frequencies of these potentially inflammatory cells (Thy1.2/CD90.2$^+$CD3$^+$CD4$^+$CD44$^+$) were restrained across lymphoid and gut tissues of recipients of both isolated Tregs and lentiviral *Foxp3*-expressing cells. In contrast, expressers of the K262R mutant *Foxp3* were profoundly unable to control effector cell numbers or frequencies in recipient mice in this model (Appendix Fig S4A–C). Reinforcing the notion that modification at K262 is important for stable FOXP3 expression, across the tissues involved, recipients of the K262R mutant saw marked loss of FOXP3 staining in the injected (Thy1.1/CD90.1$^+$) "Treg" population. In contrast, most wild-type *Foxp3* transductants recovered in this experiment still expressed FOXP3 (Appendix Fig S4D and F). These results further suggest that ubiquitination of FOXP3 at K262 is important for Treg function and stability. Moreover, preventing ubiquitination at K262 resulted in a failure to suppress production of the proinflammatory cytokines IFN-γ and IL-17 by T cells (Appendix Fig S4G and H)—an observation supporting the notion that an effective and functionally stable suppressor population depends heavily upon K63-type ubiquitination of FOXP3.

# Discussion

Sustained FOXP3 expression is a defining characteristic of Tregs, and it is necessary for their ability to maintain immunological self-tolerance through suppressive functions. However, Tregs are subject to a number of stabilizing and destabilizing stimuli that can impact their function through posttranslational modifications made to FOXP3. Previously, we and our colleagues uncovered a pathway for the degradation of FOXP3 protein hinging on K48-type polyubiquitination (Chen *et al*, 2013; van Loosdregt *et al*, 2013). In the current study, we characterize a distinct, non-proteolytic pathway of K63-type ubiquitination mediated by TRAF6 that is responsible for enforcing the proper trafficking and function of FOXP3.

TRAF6 is an E3 ligase well known for mediating K63-type ubiquitination and a widely studied member of the tumor necrosis factor (TNF)-associated factor family of adaptor molecules. It participates in a number of signaling pathways triggered by the TLR/IL-1 family of receptors as well as other TNF receptor family members. These pathways can be crucial for the activation of transcription factors including NFκB and AP-1 (Inoue *et al*, 2007). Despite this association with pathways critical to immune activation, many studies have revealed a role for the E3 ligase in immune control. For instance, TRAF6-deficient mice are prone to autoimmunity (Chiffoleau *et al*, 2003; Akiyama *et al*, 2005) suggesting a key role in the regulation of immune activation.

TRAF6 has also been specifically described as important for Tregs. Genetic deletion of this ligase reduces the frequency of FOXP3-expressing single positive CD4$^+$ in the thymus (Shimo *et al*, 2011). Since the molecules involved in TCR-triggered NFκB activation (i.e., CARMA1, c-Rel) have been shown to be necessary for thymic Treg development (Barnes *et al*, 2009; Long *et al*, 2009; Ruan *et al*, 2009), a positive role for TRAF6 in this initial upregulation of FOXP3 was proposed. Moreover, Treg frequencies in the peripheral lymphoid tissues of global TRAF6 knockout mice were also reported to be reduced compared to wild-type mice. Curiously though, *in vitro* induction of FOXP3 expression in T cells by TGFβ was found to be more robust in the absence of TRAF6 expression (Cejas *et al*, 2010; Shimo *et al*, 2011). While these prior reports suggest a role for TRAF6 in the development of Tregs in the thymus, an additional role for this E3 ligase was recently identified—one ensuring the phenotypic stability of Tregs in peripheral tissues.

Previously, Muto and colleagues reported on the consequences of Treg-specific TRAF6 deficiency. In line with our own results, they found Traf6$^{fl/fl}$Foxp3Cre$^+$ mice to be prone to dermatitis, lymphoproliferation, and widespread immune pathologies. Also, they found that Tregs from these mice were only modestly less suppressive *in vitro* than their wild-type-derived counterparts. Yet, knockout Tregs were found to be ineffectual suppressors *in vivo*. Additionally, using Cre-recombinase-driven, Treg fate-tracking mice, these authors clearly showed that unlike their wild-type counterparts, Tregs lacking TRAF6 expression are phenotypically unstable, readily losing FOXP3 expression in favor of apparent effector function (Muto *et al*, 2013).

In accord with these observations, we also found that Treg-specific ablation of TRAF6 resulted in profound defects in the function of these cells *in vivo* but not *in vitro*. Potential reasons for this consistently observed *in vivo–in vitro* disconnect may be found in the partially distinct Treg "skill sets" being tested by widely used assays of suppressive function (Sakaguchi *et al*, 2009). Given our findings that TRAF6 deficiency seemed to render Tregs more susceptible to inflammatory cue-induced FOXP3 down-regulation (Fig EV1), it is tempting to speculate that the diverse and complex environmental inputs encountered *in vivo* (e.g., abundant inflammatory cytokines, microbial products, or differential access to Treg-supporting cytokines such as IL-2 and nutrients) might pose added destabilizing effects on TRAF6-deficient Tregs that they, unlike their wild-type counterparts, cannot overcome. It is worth pointing out that while genetic ablation of TRAF6 in Tregs did not impair their *in vitro* function, rendering FOXP3 insensitive to K63 ubiquitination at K262 did. This could indicate that another, cooperative yet unknown mediator of FOXP3 K63 ubiquitination or a distinct posttranslational modification at the site contributes to Treg function. It might also simply reflect a more tenuously established suppressive repertoire in the ectopic *Foxp3*-expressing "Tregs" used in our K262R characterization relative to that of *ex vivo*-isolated nTregs in our wild-type-knockout comparisons.

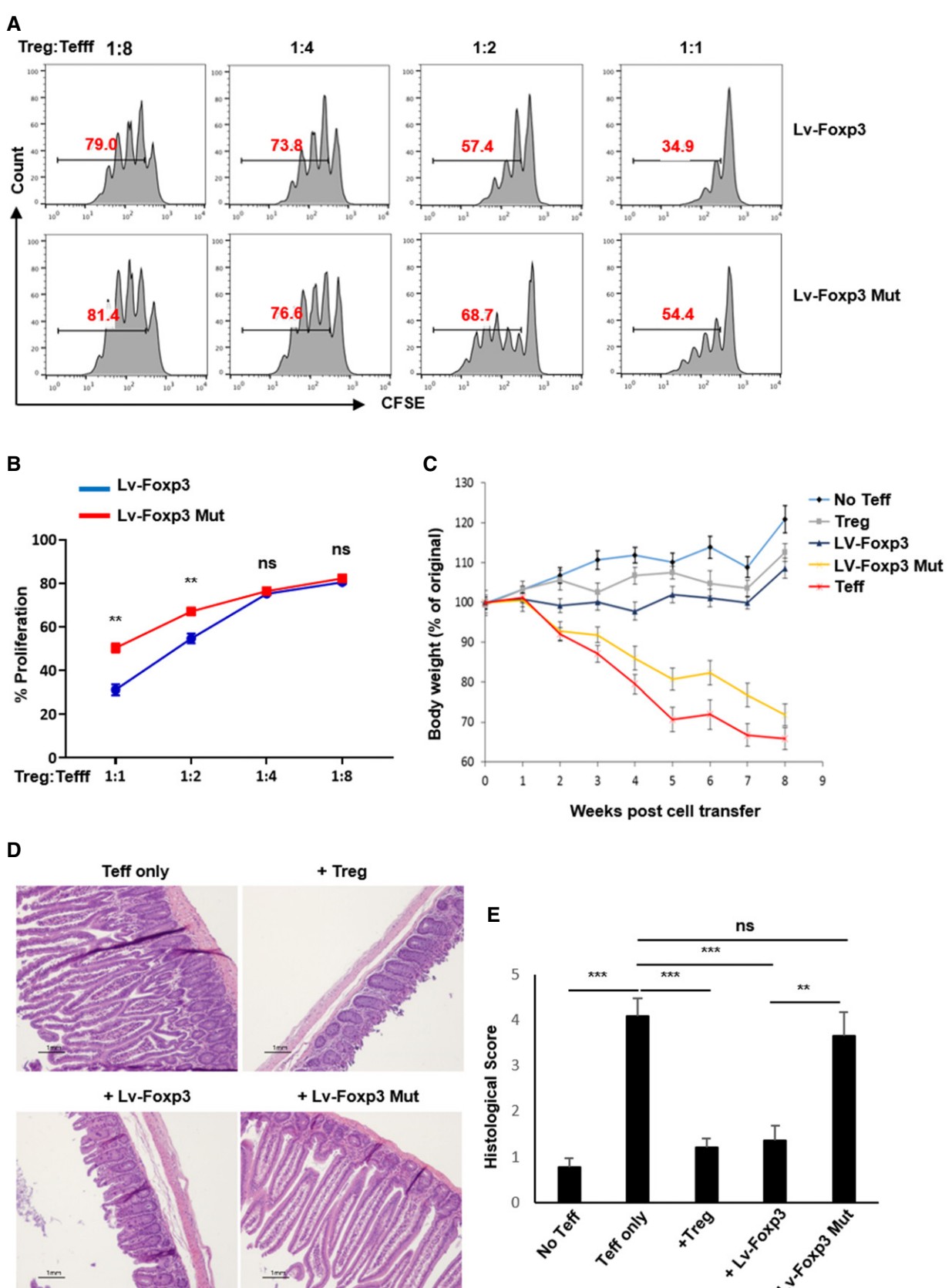

**Figure 6.**

◀

**Figure 6.   Genetic prevention of ubiquitination at Foxp3 residue K262 disrupts immune suppression *in vitro* and *in vivo*.**

A, B   The *in vitro* suppressive potency of wild-type- and K262R *Foxp3*-expressing T cells. Naïve CD4$^+$ T cells were purified from Thy1.1$^+$ BALB/c mice and subjected to retroviral transduction to express either wild-type FOXP3 or a K262R mutant resistant to ubiquitination at lysine residue 262. These engineered "Tregs" were then co-cultured with naïve responder CD4$^+$ T cells stained with CFSE at the indicated ratios and activated with anti-CD3/CD28 antibodies. Dilution of CFSE signal by the responder populations (i.e., proliferation) was assessed by flow cytometry.

C   *In vivo* suppression of colitis by wild-type- and K262R mutant-*Foxp3*-expressing T cells. $2 \times 10^5$ naïve CD4$^+$ transductants from A were mixed with $1 \times 10^6$ colitogenic naïve CD4$^+$ T cells from wild-type mice and transferred i.v. into Rag2$^{-/-}$ mice. Rag2$^{-/-}$ mice receiving no cell transfers, naïve CD4$^+$ T cells alone, or purified wild-type Tregs served as controls (*n* = 6/group). Recipient mouse body weights were monitored weekly for each group.

D, E   After 8 weeks, colons were excised, fixed in buffered formalin, and processed for H and E staining (D) and histopathology scoring (E). Scale bars shown in 1 mm).

Data information: Panels (B, C and E) represent mean values ± SEM from two experiments. **$P < 0.01$; ***$P < 0.001$; ns, no significance (unpaired *t*-test). Panel (A) and the micrographs depicted in (D) are representative findings.

Interestingly, Muto *et al* found that Traf6$^{fl/fl}$Foxp3Cre$^+$ mice harbored greater proportions and numbers of FOXP3-expressing T cells than wild-type mice in several tissues at baseline (Muto *et al*, 2013). It is possible, especially given the unstable expression of FOXP3 convincingly shown by these authors in their Treg fate-tracking experiments that a number of variables such as culture conditions and differential exposure to inflammatory cues (possibly reflecting differing animal facility conditions, commensal microbial composition, etc.) might impact FOXP3 expression among the phenotypically fragile TRAF6-deficient Treg pool. Aside from this observation, our present findings largely confirm and expand upon those of Muto *et al* as they resoundingly demonstrate that without TRAF6, FOXP3 protein levels and the gene expression pattern typical of Tregs are poorly maintained to the detriment of immune control at baseline and in distinct disease models (i.e., colitis and cancer). Beyond confirming the importance of TRAF6 as a stabilizer of the Treg phenotype and adding to the list of scenarios wherein TRAF6 is critical for the enforcement of tolerance, the present study identifies the mechanism underlying this molecule's role in Tregs. Specifically, we reveal that this E3 ligase facilitates a previously unknown mode of posttranslational control over FOXP3 expression and function. We furthermore present a characterization of this K63-type ubiquitination-dependent mechanism contributing to the proper cellular distribution of this central molecular player in Treg gene expression and function.

We and our colleagues also recently suggested a positive relationship between TRAF6 expression and Treg function. Particularly, we found that by antagonizing a specific microRNA (miR-146b), which targets TRAF6 transcript, Foxp3 protein levels could be enhanced. Moreover, the *in vitro* and *in vivo* function of Tregs could be bolstered through treatment with a miR-146b antagomir (Lu *et al*, 2016). It should be pointed out that this prior study attributed increased FOXP3 expression and Treg function to elevated NFκB activity, which can drive transcription of the *Foxp3* gene (Chiffoleau *et al*, 2003; Long *et al*, 2009), but levels of *Foxp3* mRNA were not specifically measured. In light of our current findings, it seems likely that TRAF6 may contribute to both transcriptional and protein-level regulation of FOXP3 and Treg function. The possibility of multiple roles for TRAF6 in Treg biology begs further study.

Ligase-independent functions have been attributed to TRAF6. In fact, studies utilizing an enzymatically defective, TRAF6 (L7H4) knock-in mouse revealed that the widely recognized ability of this factor to act as an E3 ligase was not necessary for a subset of the gene expression changes triggered in macrophages by TLR/MYD88 signaling. Other cellular responses governed by

this signaling pathway were, in contrast, completely prevented by a lack of TRAF6 ligase activity (Strickson *et al*, 2017). As this signaling machinery can have major impact on the response of Tregs to their surrounds in health and disease settings, the potential contributions of these and other potential ligase-independent activities of TRAF6 to Treg biology should be explored. Indeed, the relative importance of TRAF6's ligase-independent and ligase-dependent functions in FOXP3 regulation and Treg function is the subject of ongoing research efforts.

Much has been elucidated regarding the transcriptional control of FOXP3 expression. For instance, transcription at the *Foxp3* gene can be dictated by epigenetic mechanisms and the interaction of multiple transcription factors (including FOXP3 itself) with several regulatory conserved non-coding regions (CNS1, CNS2, CNS3), each with unique functions (Lu *et al*, 2014). Additionally, it is now appreciated that FOXP3 function is largely facilitated by a number of co-regulator molecules such as Eos, IRF-4, GATA-3, and other members of the Foxp3 "interactome" (Pan *et al*, 2009; Fu *et al*, 2012; Hori, 2012; Lu *et al*, 2017). Posttranslational modifications capable of influencing FOXP3 expression and activity are just now being brought to light (Li *et al*, 2015). Surprisingly, given the undisputed importance of FOXP3 and Tregs for the maintenance of immune homeostasis, relatively little is known concerning how FOXP3 protein distribution is controlled in Tregs. The present study reveals a previously unforeseen mechanism ensuring the accumulation of FOXP3 in the nucleus that hinges on the posttranslational modification of FOXP3 protein by TRAF6.

We have shown that this regulation depends upon a specific K63-type ubiquitination event at the FOXP3 lysine residue 262, which is facilitated by TRAF6. It is worth noting that this same residue (or its human equivalent, K263) is among those lysine residues previously implicated as a site for both stabilizing acetylation (Kwon *et al*, 2012) and degradative ubiquitination events on FOXP3 protein (Chen *et al*, 2013; van Loosdregt *et al*, 2013). It is possible, in light of our present findings, that K63 modification at this site competes with K48-type ubiquitination events that can direct the proteasomal turnover of FOXP3. Even more likely, however, is the possibility that K63 ubiquitination at residue K262 can render FOXP3 protein less susceptible to K48-ubiquitination events at other sites, which may otherwise drive proteasomal turnover of FOXP3. Cross-talk of this kind has been suggested between FOXP3 phosphorylation events with opposing effects on the protein's activity (Li *et al*, 2014). Such an effect of K63 ubiquitination on other ubiquitin modifications could stem from alterations in FOXP3 conformation or a localization-

dependent sequestration of FOXP3 from the action of degradation promoting K48-ubiquitin ligases in the cytoplasm. Either would be in line with our finding that the perinuclear accumulation of the transcription factor seen in the absence of TRAF6 expression or activity is associated with a destabilized FOXP3 protein pool. On the other hand, it is also possible that nuclear exclusion resulting from impaired K63-type ubiquitination may deprive FOXP3 of acetylation events at a number of other lysine residues. As this type of posttranslational modification is known to enhance both the stability and chromatin-associating activity of FOXP3 (Samanta *et al*, 2008; van Loosdregt *et al*, 2010), curtailing them would be expected to adversely affect FOXP3 expression as well as function. While acetylated FOXP3 species were noticeably reduced in Tregs lacking TRAF6 expression in our studies (data not shown), suggesting some cross-talk, direct or indirect, exists between these posttranslational FOXP3 regulating mechanisms, further dissection of this potential interplay is needed.

This ubiquitin-mediated control over FOXP3's cellular distribution and activity represents yet another opportunity for novel therapies aimed at modulating immune control. Our results present a compelling case for TRAF6 as a potential therapeutic target in anti-cancer immunotherapies. Given that TRAF6 deficiency in Tregs effectively prevented the establishment of tumor-enforced immune tolerance, pharmacological targeting of TRAF6 may be a potent means to improve anti-tumor immunity. While the potential drawbacks of such an approach could include the unintended stifling of anti-tumor effector cell activation, our findings seem to dispel this concern. Specifically, deleting TRAF6 across all T cells (using Traf6$^{fl/fl}$ CD4Cre$^+$ mice) did not adversely affect the mounting of a robust anti-tumor response, and these mice, in fact, largely phenocopy Treg-specific TRAF6 knockout mice (data not shown)—a finding in line with the notion that TRAF6 is not necessary for effector T-cell responses, but is critical for Treg-mediated immune restraint. In light of this and our other findings, the development of therapeutic TRAF6 inhibitors may yield highly effective agents for sabotaging this posttranslational mode of regulating FOXP3 activity and fully unleashing the anti-tumor immune response. Yet as TRAF6 is known to play a role in other immune cells (e.g., Dendritic Cells, macrophages) and signaling pathways (e.g., NFκB activation, the PI3K cascade) with anti-tumor potential, further study is absolutely needed to test this notion and determine the relative impact of TRAF6 targeting on Treg and effector leukocyte activity in the cancer setting.

In all, these findings reveal TRAF6 as a previously unappreciated posttranslational mediator of FOXP3 expression and activity in Tregs. They also provide insights into the molecular details of this distinct, non-proteolytic, and functionally enhancing brand of ubiquitination by TRAF6 including the specific modification site in the FOXP3 molecule. Importantly, this newly uncovered mode of control over Treg function may represent a target for future immunotherapies aiming to overcome immune suppression in cancer patients.

# Materials and Methods

## Mice

Mice with a floxed *Traf6* gene (Traf6$^{fl/fl}$ mice) on a C57BL/6 background were obtained from Dr. M. Pasparakis (Polykratis *et al*,

2012). Foxp3yfpCre transgenic mice (originally generated in the laboratory of Dr. Alexander Rudensky) and CD4Cre transgenic mice were obtained from the Jackson Laboratory. These mice were crossed to generate mice specifically lacking TRAF6 in their Tregs and T cells, respectively (Traf6$^{fl/fl}$ Foxp3/CD4Cre$^+$ mice) as well as wild-type littermate controls (Traf6$^{fl/fl}$ Foxp3/CD4Cre$^-$ mice). Congenically distinct Thy1.1$^+$ and Thy1.2$^+$ mice on a BALB/c background and CD45.1$^+$ C57BL/6 mice were purchased from the Jackson Laboratory. All mice were housed in a specific-pathogen-free facility in accordance with institutional guidelines.

## Generation of mutant Foxp3 constructs

Single lysine and targeted residue mutant in *Foxp3* constructs were generated by Site-Directed Mutagenesis Kits (Thermo Fisher Scientific).

## Cell line transfections, co-IP, and Immunoblotting

To ectopically express wild-type mutant constructs in 293T cells, we utilized tagged expression vectors as indicated, and the cells were transfected with Lipofectamine 2000. For co-IP experiments, cells were lysed in RIPA buffer containing 50 mM Tris–HCl, pH 7.4, 1% Nonidet P-40, 0.5% Nadeoxycholate, 150 mM NaCl, 1 mM EDTA, with 1 mM PMSF, 1 mM Na$_3$VO$_4$, 1 mM NaF, and protease inhibitor (Sigma), followed by immunoprecipitation with the indicated antibodies, separation by SDS–PAGE (gels contained 0.1% SDS), and analysis by Immunoblotting. For all immunoblotting experiments, proteins were diluted in loading buffer containing 2% SDS and β-mercaptoethanol (100 mM) and were briefly boiled before resolution under denaturing conditions. Where applicable, band densities indicating protein amounts were quantified using ImageJ software and values were normalized to a loading control. K63-type ubiquitination was observed by using specific antibodies and anti-K63 TUBE reagent (LifeSensors) where indicated.

## Retroviral transduction of primary T cells

Naïve T cells isolated from wild-type or Traf6$^{fl/fl}$/CD4Cre$^+$ mice were stimulated with plate-bound anti-CD3 (4 µg/ml) and soluble anti-CD28 (1 µg/ml) with 60 U/ml human recombinant IL-2 for 16 h. Activated T cells were transduced with viral supernatants (carrying empty vector MigR1, wild-type *Foxp3* construct, or mutant *Foxp3* constructs) supplemented with 60 U/ml IL-2 and 4 µg/ml polybrene, followed by centrifugation for 1 h at 1,455 *g*. Cells were cultured at 37°C with 5% CO$_2$ for an additional 48 h and sorted out for ELISA analysis. All retroviral expression plasmids (inserts were cloned into Mig-R1 vector) and packaging vectors (pCL-Eco) were obtained from GeneChem.

## T-cell isolation and Treg suppression assays

Naïve CD4$^+$ T cells from C57BL/6 mice were isolated from pooled lymph nodes and spleens by FACS as were congenically distinct Tregs (CD4$^+$/CD25$^+$) isolated from either wild-type mice or the indicated conditional knockout strain prior to combination at the indicated ratios. Naïve T cells were stained with carboxyfluorescein

diacetate succinimidyl ester (CFSE) and co-cultured with equal numbers ($2 \times 10^5$) of antigen-presenting cells (obtained from the CD4-depleted fraction) in the presence of activating anti-CD3 antibody (0.5 μg/ml). Tregs were added at the indicated ratios. Seventy-two hours poststimulation, division of naïve T cells was assesses by dilution of the CFSE signal. For *in vivo* suppression assays, $CD4^+CD25^-CD62L^{high}$ naïve T cells were sorted from $CD45.1^+$ mice and $CD4^+Yfp^+$ Tregs were isolated from wild type Foxp3-Yfp$^+$Cre mice and Traf6$^{fl/fl}$ Foxp3-Yfp$^+$Cre mice, respectively. $CD4^+CD25^-CD62L^{high}$ ($1 \times 10^6$/mice) naïve T cells and $CD4^+ Yfp^+$ ($2 \times 10^5$/mice) Tregs from Foxp3-Yfp$^+$Cre mice or Traf6$^{fl/fl}$ Foxp3-Yfp$^+$Cre mice were co-injected via the tail vein (i.v.) into B6 Rag2$^{-/-}$ immunodeficient recipients (at a 5:1 ratio). Seven days later, mice were euthanized, and spleens were dissected and placed into separately labeled tubes of completed media as mentioned. Leukocytes recovered from recipient spleens were isolated and then stained for CD4 and CD45.1 after counting cell number.

### In vitro T helper subset differentiation

Suspensions of murine leukocytes were obtained from lymph nodes and spleens. Naïve $CD4^+$ T cells were then obtained by FACS based on their $CD4^+$/$CD62L^{high}$/$CD25^-$ surface marker profile. Cells were activated in 12-well plates with completed media supplemented with stimulatory antibodies against CD3 and CD28 (1 and 2 μg/ml, respectively). For the generation of Th0 cells, cells were activated without skewing cytokines. Th1 cells were activated in the presence of IL-12 (20 ng/ml) and anti-IL-4 neutralizing antibodies (10 μg/ml). Th17 skewing was driven by inclusion of IL-6 (20 ng/ml) and anti-IL-4 and anti-IL-12 (10 μg/ml each) in the activation media of naïve $CD4^+$ T cells (RPMI supplemented with 10% heat-inactivated FBS, 1% antibiotics, 1% non-essential amino acids, 1% sodium pyruvate, 1% glutamine, and 2-mercaptoethanol). iTregs were generated by activating naïve precursors as above in the presence of IL-2 (100 U/ml) and the indicated concentrations of TGFβ.

### qRT–PCR and ELISA

RNA was isolated by a miniRNA Extraction Kit (Qiagen). The cDNA Archival Kit (Applied Biosystem) was used per the manufacturer's instruction. Triplicate reactions were run using an ABI Prism 7500. mRNA levels were determined by comparative CT method and normalized to GAPDH expression. IL-2 production from the sorted T-cell culture supernatants was measured by an IL-2 ELISA Kit (Biolegend).

### Immunofluorescence and confocal microscopy

Cells were spun down onto microscope slides at 400 *g* for 3 min at room temperature (RT) using a Cytospin (Thermo Scientific). The slides were then fixed with 4% paraformaldehyde for 15 min at room temperature (RT), then washed three times with PBS and incubated with 0.5% Triton X-100 for 5 min before washing again with PBS. The cells were then blocked with 1% BSA for 1 h at RT. FITC-labeled anti-mouse FOXP3 antibodies were diluted 1:60 with PBS and incubated with the cells for 1 h at RT. After washing the slides, DAPI was used as a nuclear counterstain. Fluorescence microscopy images were obtained using an Eclipse E800 microscope

equipped with a DS-Qi1Mc camera (Nikon) and NIS-Element AR 3.0 software. Confocal microscopy was carried out using a confocal EZ-C1 microscope (SKCC Cell Imaging Facility) with Fiji–ImageJ software.

### Mouse tumor models

The melanoma cell line (B16) and the colon carcinoma cell line (MC38) were maintained *in vitro*. For tumor challenge experiments, cells were trypsinized, washed, and resuspended in PBS. $1 \times 10^5$ tumor cells were injected subcutaneously (s.c.) into the shaved flanks of age- and sex-matched Traf6$^{fl/fl}$ Foxp3Cre$^-$ (wild type) and Traf6$^{fl/fl}$ Foxp3Cre$^+$ mice (on a C57BL/6 background). Tumor volumes were measured every 2–3 days using a digital caliper. At indicated time points postinjection, cohorts of mice were euthanized, and the leukocytes infiltrating the tumor, tumor-draining and peripheral lymph nodes, and spleen were harvested and characterized by surface and intracellular immunostaining. Tumor-infiltrating leukocytes were recovered by Percoll gradient centrifugation.

### Luciferase-based transcription activity and repression assays

The IL-2 luciferase reporter plasmid was co-transfected with a Renilla luciferase encoding plasmid into 293T cells or Jurkat T cells. The cells were lysed and analyzed using a luciferase assay, and the results were normalized to Renilla luciferase activity according to the manufacturer's protocol (Promega).

### Measurement of FOXP3 protein stability

Transfected cell lines or primary murine Treg cells were cultured *in vitro* with 5 μg/ml CHX to halt *de novo* protein synthesis. At the indicated time points post-CHX treatment, cells were harvested and lysed, and the levels of cellular FOXP3 were measured by immunoblot analysis. Band densities were determined using ImageJ software, and the relative turnover rates of the FOXP3 protein pool in experimental groups were thus determined.

### Flow cytometry and ImageStream analysis

For extracellular staining, harvested cells were washed and incubated in PBS containing 1% FBS containing fluorochrome-conjugated antibodies in a U-bottom 96-well plate. For intracellular FOXP3, staining cells were fixed and permeabilized using eBioscience FOXP3 Staining Kit prior to staining. For intracellular cytokine staining, harvested cells were re-stimulated in PMA and Ionomycin in the presence of Golgi-Plug (BD). After 5-h incubation, the cells were fixed/permeabilized (eBioscience) and incubated with antibodies against IFN-γ, IL-17, or TNF-α. For cellular proliferation, Cell Trace CFSE Cell Proliferation Kit (Invitrogen) was used per manufacturer's manual. Nuclear-FOXP3 colocalization in the Tregs of Traf6$^{fl/fl}$CD4Cre$^+$ and Traf6$^{fl/fl}$CD4Cre$^-$ (wild type) mice was assessed using ImageStream analysis (Amnis). Untouched $CD4^+$ T cells were enriched from the lymph nodes and spleens of each group by magnetic bead isolation (Life Technologies), and intracellular FOXP3 and nuclei were stained with FITC-labeled antibody (eBioscience) and DAPI, respectively. Using IDEAS ImageStream analysis

software, nuclear and perinuclear masks were generated and the frequencies of FOXP3$^+$ cells in each group falling into bins designated high and low probability for nuclear localization were determined. For a summary of antibodies used in these studies, see Appendix Table S1.

### Adoptive transfer-induced colitis model

Naïve CD4$^+$ T cells from C57BL/6 mice were isolated from pooled lymph nodes and spleens by FACS and resuspended in PBS. $1 \times 10^6$ naïve T cells were injected i.v. into lymphopenic Rag2$^{-/-}$ mice. $2 \times 10^5$ congenically distinct, freshly isolated Tregs or CD4$^+$ T cells transduced with wild-type or mutant Foxp3 constructs were co-injected as indicated. Changes in body weight were assessed weekly, and upon conclusion of the experiment, colons were removed and fixed in 10% formalin. Five-micrometer paraffin-embedded sections were cut and stained with hematoxylin and eosin (H&E). The pathology of colon tissue was scored in a blinded fashion, on a scale of 0–5 where a grade of 0 was given when there were no changes observed. Changes associated with other grades were as follows: grade 1, minimal scattered mucosal inflammatory cell infiltrates, with or without minimal epithelial hyperplasia; grade 2, mild scattered to diffuse inflammatory cell infiltrates, sometimes extending into the submucosa and associated with erosions, with mild to moderate epithelial hyperplasia and mild to moderate mucin depletion from goblet cells; grade 3, moderate inflammatory cell infiltrates that were sometimes transmural, with moderate to severe epithelial hyperplasia and mucin depletion; grade 4, marked inflammatory cell infiltrates that were often transmural and associated with crypt abscesses and occasional ulceration, with marked epithelial hyperplasia, mucin depletion; and grade 5, marked transmural inflammation with severe ulceration and loss of intestinal glands. Transferred naïve and Treg cell populations were characterized by surface marker and intracellular staining for cytokines and Foxp3. Leukocytes recovered from recipient lymph node, spleen, and lamina propria were re-stimulated before surface staining for CD4, CD45.1, and CD45.2.

### Statistics

The significance of differences was determined by an unpaired Student's *t*-test, one-way ANOVA, and two-way ANOVA using Prism software (GraphPad). Differences were considered significant when *P* values were < 0.05.

**Expanded View** for this article is available online.

### Acknowledgements

L.L.'s research is supported by the National Natural Science Fund of China (grants 81571564, grant 91442117, grants 81521004 and 81522020), the 863 Young Scientists Special Fund (grant SS2015AA020932), National Science Foundation of Jiangsu Province BK20131024, BE2016766. We thank H. K. Lin (Wake Forest University School of Medicine, USA) for valuable reagents. F.P.'s research was supported by grants from the Bloomberg-Kimmel Institute, the Melanoma Research Alliance, the National Institutes of Health (R01AI099300, R01AI089830, R01AI137046, and R01CA218270), Department of Defense (PC130767), Emerson Collective Award, and Johns Hopkins Discovery Award. F.P. is a Stewart Trust Scholar; J.B.'s research is supported by a grant from the Roswell Park Alliance Foundation and NCI grant P30CA016056.

### Author contributions

XN, JB, FP, and LL designed the experiments; XN, WK, PW, JT, JG, BVP, ZC, SN, and JB performed experiments; XN, FP, and LL analyzed data; HS, XW, BL, and BRB gave suggestions and critical discussion; XN, JB, and FP wrote the paper; XN, WK, PW, XW, JF, HP, WY, YT, XY, SN, AL, and XW revised the manuscript.

### Conflict of interest

The authors declare that they have no conflict of interest.

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
