## [Review Process File · The EMBO Journal]

TRAF6 directs FOXP3 localization and facilitates regulatory T cells function through K63-linked ubiquitination

Xuhao Ni, Wei Kou, Jian Gu, Ping Wei, Xiao Wu, Hao Peng, Jinhui Tao, Wei Yan, Xiaoping Yang, Andriana Lebid, Benjamin V. Park, Zuojia Chen, Yizhu Tian, Juan Fu, Stephanie Newman, Xiaoming Wang, Hongbin Shen, Bin Li, Bruce R. Blazar, Xuehao Wang, Joseph Barbi, Fan Pan and Ling Lu.

Review timeline:

Submission date:	4 th May 2018
Editorial Decision:	29 th June 2018
Revision received:	5 th November 2018
Editorial Decision:	14 th January 2019
Revision received:	25 th January 2019
Accepted:	15 th February 2019

Editor: Elisabetta Argenzio

Transaction Report:

1st Editorial Decision

29th June 2018

Thank you for submitting your manuscript on TRAF6-mediated FOXP3 ubiquitination and Treg regulation to The EMBO Journal. We have now received three referee reports on your study, which are enclosed below for your information.

As you can see, all the referees find TRAF6-mediated, K63-linked polyubiquitination of FOXP3 novel and interesting. However, they also raise several critical points that need to be addressed before they can support publication at The EMBO Journal. In particular, referee #1 and #3 request you to test FOXP3 ubiquitination under denaturing conditions and to clarify the role of TRAF6-induced acetylation of FOXP3. In addition, they find that reduced K63 ubiquitination in Traf6^{-/-} Treg cells needs to be further investigated (e.g. reconstituting Traf6^{-/-} cells with either wt TRAF6 or mutants with disrupted E3 ligase activity). Referee #1 also states that both the discrepancy between Traf6^{-/-} Treg immunosuppressive properties in vivo versus in vitro, as well as the role of ubiquitination for FOXP3 localization versus stabilization need to be addressed. Finally, referee #3 requests a more immunogenic mouse model be employed to show the effect of Treg deficiency.

Addressing these issues through decisive additional data as suggested by the referees would be essential to warrant publication in The EMBO Journal. Given the overall interest of your study, I would thus like to invite you to revise the manuscript in response to the referee reports.

REFeree REPORTS

Referee #1:

The study by Ni and colleagues identifies a novel role for TRAF6 in regulating regulatory T cell function through the K63 linked polyubiquitination of the T reg transcription factor FOXP3. The

authors propose that TRAF6 mediates the K63 linked polyubiquitination at K262 of FOXP3 which promotes the nuclear localisation of FOXP3. A lack of FOXP3 localisation to the nucleus reduces the immunosuppressive properties of T regs *in vivo*, but not *in vitro*. This leads to enhanced tumour suppression in mice lacking TRAF6 in T regs and a failure of T regs overexpressing a FOXP3 K262R mutant to suppress T cell dependent colitis. The data is generally of high quality and the study incorporates both molecular, cellular and physiological measures of FOXP3 function and will significantly contribute to our understanding of TRAF6, FOXP3 and T reg cells. There are some issues however that should be addressed

Major points

- The data presented in figure 3 are poorly labelled and as such are difficult to interpret.
 - o In Fig 3A what antibody is used to immunoblot? The expression levels of TRAF6 (WT and C70A) and FOXP3 should be shown. It is not clear how this data matches the description provided in the corresponding figure legend.
 - o In Fig 3B what is the third lane? The expression levels of immunoprecipitated TRAF6 as well as the levels of FOXP3 and TRAF6 in the inputs used for IP should be shown.
 - o In figure 3C what cells are used?
 - o The data in figure 4D appears to be mislabelled, presumably each IP was probed with both anti-FOXP3 and anti-TRAF6 antibody?
- The data presented in figure 3e is not sufficient to demonstrate reduced K63 polyubiquitination of FOXP3 in *Traf6*^{-/-} cells. The levels of total FOXP3 protein immunoprecipitated in each lane needs to be shown to ensure equal levels of FOXP3 are analysed in each sample. The levels of TRAF6 in the cell lysates should also be shown. From the levels of FOXP3 shown in the inputs it seems likely that uneven levels of FOXP3 in the lysates used for immunoprecipitation underlie the reduced K63 polyubiquitination seen. It is also not clear why the acetylated form of FOXP3 is measured in the immunoprecipitations and its relevance is not mentioned in the main text. This data suggests that TRAF6 also controls FOXP3 acetylation but this is not followed up by the authors.
- The measurements of ubiquitination should be performed in whole cell lysates which have been denatured to ensure that the ubiquitination seen is of FOXP3 protein and not an interacting protein. It is not clear from the description of the experimental methods what conditions have been used in this study.
- The authors identify K262 as a target residue for TRAF6 mediated ubiquitination of FOXP3. Does mutation of K262 also block FOXP3 interaction with TRAF6? The supplementary data shown in SFig 5 measures TRAF6 and FOXP3 co-localisation not interaction. In addition, K263 of FOXP3 has previously been demonstrated to be K48 polyubiquitinated but it has not been tested here. Is K263 also ubiquitinated by TRAF6?
- The loss of TRAF6 E3 ligase activity would be expected to recapitulate the phenotype of the FOXP3 K262 mutant. Reconstitution of *Traf6*^{-/-} cells with WT or C70A TRAF6 would allow this to be tested.
- The authors data shows that *Traf6*^{-/-} T regs retain immunosuppressive properties *in vitro* but are defective in immunosuppression *in vivo*. How do the authors explain this difference? How does this relate to the proposed mechanism of TRAF6 mediated regulation of FOXP3 which should be cell intrinsic?
- The immunofluorescence microscopy data shown in figure f and supplementary figure 6 is not of sufficiently high quality to justify the authors conclusions. Higher magnification confocal microscopy images should be included to more convincingly demonstrate the altered localisation of FOXP3 in *Traf6*^{-/-} T regs.
- The authors should clarify if the WT controls indicated in Figure 2, figure 5 and figure Sf6 are wild type C57/B6, or *Foxp3* Cre mice which are more appropriate controls.
- The data shown in Sfig 7 suggests that FOXP3 K262 mutation reduces FOXP3 stability *in vivo*. It is not clear how this fits with the proposed mechanism of TRAF6 mediated K63 ubiquitination which should alter subcellular localisation rather than stability.

Minor points

- To avoid confusion the authors should use the correct nomenclature when referring to FOXP3 and TRAF6 proteins and *Foxp3* and *Traf6* (in italics) genes, particularly since the study incorporates measures of both.
- No information on antibody sources and clones used is provided in the methods section. Over all the methods section lacks the appropriate level of detail required to allow replication of the experiments described.

- There are a number of typographical errors in the labelling of figures which may not be picked up by copy editors.

Referee #2:

Ni et al studied the functions of TRAF6 deletion in Foxp3⁺ Treg cells. TRAF6 deletion impaired Treg function, and TRAF6 promoted K63 ubiquitination at lysine 262. The latter event was shown to be required for optimal Foxp3 function.

Main comment

There is already quite a literature on TRAF6 being required for optimal Treg function, including references 17-23 cited in the paper and others, and so the new part of the data is that relating to K63 ubiquitination of Foxp3. I.e. Given prior studies of the effects of conditional deletion of TRAF6 in Foxp3⁺ Tregs I don't think Figures 1 and 2 add much to the paper, and these Figures could well be deleted or simply consigned to the supplement. In contrast, the new material relating to the K63 ubiquitination of Foxp3 is of significant interest.

Minor comments

1. As it stands, the authors should document the purity of the cell populations under study in Figure 1; i.e. iTreg development is never 100% so what proportion of cells were Foxp3⁺ in panel a, and likewise, what proportion of the cells in panel b were Foxp3⁺?
2. In Fig. 3, panel B, what is shown in lane 3 (lysate?), and where is the expected reciprocal pull-down (IP HA and IB Flag)?
3. Some of the text simply goes whacky in places, e.g. What to make of the last sentence of the first paragraph of the Discussion? Likewise, the sixth paragraph of the Discussion needs attention ("we recently suggested positive relationship between TRAF6 expression and Treg function").

Referee #3:

Overall this is an interesting study that increases our understanding of the posttranslational mechanisms that regulate Foxp3 activity and thus Treg function. Although the first part of the study is not novel as mice deficient for TRAF6 in Treg have been described before and TRAF6 was shown to be essential for Treg functional activity *in vivo* (ref 22), the data on TRAF6-mediated K63 ubiquitination of Foxp3 are novel. The authors show that TRAF6 is able to add K63-linked polyubiquitin to FoxP3 specifically at residue K262, and that this ubiquitination is required for nuclear translocation and FoxP3 function. The authors also nicely show that this ubiquitination event is important *in vivo* by a Teff transfer model for colitis, where overexpression of wild-type FoxP3 in the transferred T cells by lentiviral transduction induces protection, but the K262R mutant does not. The reduced Treg functionality in TRAF6-deficient Treg mice also resulted in greater immune responses and anti-cancer activity in a B16 tumor model. The authors suggest that targeting TRAF6 could have potential to break tolerance and increase anti-tumor responses. Some issues still need to be addressed.

Major comments:

- Fig. 3: There are several issues with this figure. 1. the legend should better describe how the experiment was performed and what exactly is shown. 2. How can the authors exclude that the ubiquitination signal in their Co-IP experiments is coming from FOXP3 and not from another co-IPed protein. 3. Fig. 3D: labelling of the lower 2 panels is wrong. This experiment can also benefit from a similar set-up using TRAF6 KO cells as negative control to prove specificity of the antibodies used. 4. Why do the authors use ubiquitin mutants to make their point for K63-Ub in the

case of Hek293 cells, while using K63-specific antibodies in the case of primary cells. K63-specific antibodies should also be used in case of the cell line. 5. Why do the authors show acetylated FOXP3 (mentioned in legend, while no reference to K63-UB is made).

- The authors map TRAF6 binding to the zinc finger domain in FoxP3 (Figure S4), but do not show whether TRAF6 binding is required for FoxP3 ubiquitination (for example by comparing TRAF6-mediated ubiquitination of the FoxP3 mapping clones C2 vs C3 from figure S4).

- The TRAF6-C70A point mutant (Fig 3A) causes more extensive structural changes to TRAF6 than only disrupting its E3 ligase activity. A good alternative for inactive TRAF6 could be the L74H, which disrupts E3 ligase activity (PMID: 28404732). Moreover, this published paper describing a ligase-inactive TRAF6 knock-in mouse also indicates ligase-independent roles of TRAF6 (which are also disrupted by C70A). The latter should be discussed by the authors in view of their own results and might be important for the interpretation of the Treg-specific TRAF6 KO data.

- Could redundancy explain why TRAF6-deficient Tregs are possible to generate in vitro?

- Why do the authors use the B16 model which is known to be weakly immunogenic? Use of another more immunogenic mouse model seems more relevant to show the effect of Treg deficiency.

- The writing style should be much improved and the manuscript needs to be revised by a native English speaking person.

Minor comments:

Abstract: rephrase the sentence "... perinuclear accumulation, disrupted function ..."

Figure labelling in text should be corrected (Fig.Fig should be Fig.)

Fig 1D : the truncated scale for the spleen cell count could lead to interpretations of a larger difference than there actually is.

Page 10: the text mentions effects of K63-Ub at K262 on FOXP3 stability, while the data only show effects on nuclear localisation

Page 12, discussion. Also the lineage tracing experiment that shows loss of FoxP3 expression in TRAF6-deficient Tregs (effectively reverting them back to Teff cells) should be mentioned here (PMID: 24058613). This should not be dependent on sampling time as mentioned in the current discussion.

The authors make the statement that potential therapies targeting TRAF6 may be highly effective at breaking tolerance. This is an overstatement, knowing the many other functions of TRAF6.

Reference 22 should be corrected (show full name of last author)

Details and source of plasmids + antibodies used should be provided.

Point-by-point responses to Reviewer Concerns:

Referee #1:

The study by Ni and colleagues identifies a novel role for TRAF6 in regulating regulatory T cell function through the K63 linked polyubiquitination of the Tregs transcription factor FOXP3. The authors propose that TRAF6 mediates the K63 linked polyubiquitination at K262 of FOXP3 which promotes the nuclear localization of FOXP3. A lack of FOXP3 localization to the nucleus reduces the immunosuppressive properties of Tregs in vivo, but not in vitro. This leads to enhanced tumour suppression in mice lacking TRAF6 in Tregs and a failure of Tregs overexpressing a FOXP3 K262R mutant to suppress T cell dependent colitis. The data is generally of high quality and the study incorporates both molecular, cellular and physiological measures of FOXP3 function and will significantly contribute to our understanding of TRAF6, FOXP3 and T reg cells. There are some issues however that should be addressed

We thank the Reviewer for their kind comments and assessment of our findings' significance.

Major points

- The data presented in figure 3 are poorly labelled and as such are difficult to interpret.

In our revised manuscript, we have made extensive changes to the labeling and accompanying figure legend for this figure. We apologize for the lack of clarity.

- o In Fig 3A what antibody is used to immunoblot? The expression levels of TRAF6 (WT and C70A) and FOXP3 should be shown. It is not clear how this data matches the description provided in the corresponding figure legend.

We apologize for the oversight. We now more clearly indicate in the figure labels which antibodies were used to immunoprecipitate (IP) and immunoblot (IB) wherever applicable. The corresponding legend has been revised as well. In our revised Fig. 3A, we now document the largely comparable expression levels of FOXP3 in all samples as well as that of wild type and TRAF6 mutants in whole cell lysates.

- o In Fig 3B what is the third lane? The expression levels of immunoprecipitated TRAF6 as well as the levels of FOXP3 and TRAF6 in the inputs used for IP should be shown.

In Fig 3B, the 3rd lane depicts the input control. At the Reviewer's suggestion, we now include measurements of immunoprecipitated TRAF6 and FOXP3 protein in the whole cell lysate used as input in these experiments.

- o In figure 3C what cells are used?

The original Figure 3C depicted an experiment done with HEK293T cells. We now clarify this in the revised figure legend and text.

- o The data in figure 4D appears to be mislabelled, presumably each IP was probed with both anti-FOXP3 and anti-TRAF6 antibody?

Thank you for spotting the erroneously duplicated labeling in the original Fig. 3D. As you rightly suspected, each set of IP experiments involved the pull-down of proteins with either anti-FOXP3 or anti-TRAF6 antibodies followed by probing with antibodies against the reciprocal target. Both antibody sets used are now correctly indicated in our revised Fig. 3D

- The data presented in figure 3e is not sufficient to demonstrate reduced K65 polyubiquitination of FOXP3 in Traf6^{-/-} cells. The levels of total FOXP3 protein immunoprecipitated in each lane needs to be shown to ensure equal levels of FOXP3 are analysed in each sample. The levels of TRAF6 in the cell lysates should also be shown. From the levels of FOXP3 shown in the inputs it seems likely

that uneven levels of FOXP3 in the lysates used for immunoprecipitation underlie the reduced K63 polyubiquitination seen. It is also not clear why the acetylated form of FOXP3 is measured in the immunoprecipitations and its relevance is not mentioned in the main text. This data suggests that TRAF6 also controls FOXP3 acetylation but this is not followed up by the authors.

The Reviewer raises multiple good points. We have repeated the experiment in question from Fig. 3E while verifying that total FOXP3 levels in the loading material are indeed comparable in each lane. The revised Fig. 3E depicts the levels of FOXP3 and actin (loading control) in the whole cell lysate used for IP. We now also measure all FOXP3 within the K63- ubiquitinated protein fraction (proteins immunoprecipitated by anti-K63) as well as all K63-ubiquitinated proteins with the total FOXP3 protein pool (obtained by IP using anti-FOXP3 antibody). As the TRAF6 expression level in these cells is pre-determined by their donor genotype, and this conditional knockout strategy is well established, measurement of immunoprecipitated TRAF6 is not necessary.

Regarding FOXP3 acetylation: We apologize for presenting this piece of data without sufficient context. In short, this type of post-translational modification is linked to enhanced FOXP3 protein stability and function (van Loosdregt, et al. Blood. 2010 Feb 4;115(5):965-74. PMID:19996091 ; Samanta, et al. Proc Natl Acad Sci U S A. 2008 Sep 16;105(37):14023-7. PMID:18779564). It has also been suggested that degradative ubiquitination (K48-Ubiq) may compete with this stabilizing modification due to potentially shared lysine residue targets in FOXP3 molecules (Kwon et al. J Immunol 2012). The molecular details of the interplay between FOXP3 K63 Ubiquitination, nuclear localization, and its K48-Ubiq mediated degradation is an active subject of investigation that may be complex and likely falls outside the scope of the current study.

Accordingly, we have removed this result from our data set for clarity, and we now only mention it in our Discussion section for its possible relevance to the potential interplay between different posttranslational modifications made to FOXP3.

- The measurements of ubiquitination should be performed in whole cell lysates which have been denatured to ensure that the ubiquitination seen is of FOXP3 protein and not an interacting protein. It is not clear from the description of the experimental methods what conditions have been used in this study.

We thank the Reviewer for bringing up this important point. In fact, our Immunoblot measurements of ubiquitinated FOXP3 were carried out using denaturing conditions. Resolution of whole cell lysates and immunoprecipitated fractions thereof was achieved under conditions meant to denature and dissociate protein complexes. Before running samples on protein SDS-PAGE gels (typically running gels contained 0.1% SDS), they were routinely diluted in loading buffer containing 2-mercaptoethanol (100mM) and 2% SDS and were briefly boiled. We apologize for omitting these details from our original manuscript.

As the direct measurement of ubiquitination in whole cell lysates (without selective pull-down of specific factors before hand) would not likely provide information specifically relevant to FOXP3 modification, we have chosen instead to examine K63 ubiquitination of this particular protein using an IP as a key step in our approach.

- The authors identify K262 as a target residue for TRAF6 mediated ubiquitination of FOXP3. Does mutation of K262 also block FOXP3 interaction with TRAF6? The supplementary data shown in SFig 5 measures TRAF6 and FOXP3 co-localisation not interaction. In addition, K263 of FOXP3 has previously been demonstrated to be K48 polyubiquitinated but it has not been tested here. Is K263 also ubiquitinated by TRAF6?

We thank the Reviewer for their insightful suggestions. We have augmented the colocalization data in our original Supplemental Figure S5 with a reciprocal co-IP data showing the interaction of TRAF6 with both Wild type FOXP3 and the K262R mutant. As shown in the new Figure EV4B, the FOXP3 mutant in question is still able to interact with TRAF6 suggesting that the defects observed for the 262R variant likely reflect the disrupted enzymatic targeting of a key lysine residue, rather than an interaction defect.

As the Reviewer indicates, the lysine residue we implicate as being modified by TRAF6 (numbered K262 in mice, K263 in humans) has indeed been demonstrated to be a target for the apparently functionally opposite, degradation-promoting, process of K48-Ubiquitination along with other lysine residues in FOXP3. It is possible that the processes of K63- and K48-Ubiquitination actively compete for target substrates – a relationship already proposed for Foxp3 stabilizing acetylation events and K48-Ubiquitination. It seems likely, given our results, that in the absence of what appears to be a very specific site for K63-Ubiquitination, K48-modifications can be enhanced at other sites, driving accelerated protein turn-over of FOXP3 that we observe in association with aberrant cellular distribution. Whether K63-Ubiquitination at K262 directly inhibits K48-Ubiquitination elsewhere, in a manner similar to certain phosphorylation events in FOXP3 (Li, Z. et al. JBC 2014, 289, 26872-26881), or if the cytoplasmic accumulation of FOXP3 molecules lacking K63-ubiq. modification renders them more susceptible to K48-ubiquitinating enzymes remains to be explored. While potentially complex enough for a paper of its own and outside the scope of the current study, we speculate on these potentialities in our revised Discussion section.

- The loss of TRAF6 E3 ligase activity would be expected to recapitulate the phenotype of the FOXP3 K262 mutant. Reconstitution of Traf6^{-/-} cells with WT or C70A TRAF6 would allow this to be tested.

This is an excellent suggestion. We have reconstituted murine CD4⁺ T cells lacking TRAF6 (from TRAF6 fl/fl CD4Cre⁺ mice) with retroviral expression vectors encoding FOXP3 and either Wild Type TRAF6 or the mutants L74H and C70A. Indeed we found that while delivery of wild type TRAF6 could effectively rescue K63-Ubiquitination of FOXP3 and the regulatory activity of the transcription factor (as evidenced by relative expression/repression of IL-2) in these cells, the ligase-defective mutant was unable to do so, recapitulating our observations made using the K262 mutant. These results are now shown in the new Figure EV4C and D) of the revised paper.

- The authors data shows that Traf6^{-/-} T regs retain immunosuppressive properties in vitro but are defective in immunosuppression in vivo. How do the authors explain this difference? How does this relate to the proposed mechanism of TRAF6 mediated regulation of FOXP3 which should be cell intrinsic?

The Reviewer raises an interesting point. Both we and Muto et al. (PLoS One. 2013 Sep 13;8(9):e74639) find that Tregs lacking TRAF6 expression, are profoundly unable to enforce immune tolerance in different in vivo disease models as well as in unchallenged (naïve) mice. Despite this, both our study and theirs found that these same Tregs were competent suppressors in the widely used in vitro Treg suppression assay. This consistent in vitro/in vivo disparity is not unprecedented as genetic ablation or blockade of a number of factors besides TRAF6 have been reported to have little-to-no impact on in vitro Treg function while drastically impacting their in vivo function and immune control.

In fact, in vitro suppression assays have been shown to primarily test a set of Treg suppressive mechanisms (i.e. growth factor sequestration) partially distinct from those required to for in vivo function (reviewed by Sakaguchi, et al. International Immunology, Volume 21, Issue 10, 1 October 2009, Pages 1105–1111) whereas in vivo suppression (e.g. in vivo suppression assays, the adoptive transfer colitis model) appears to depend upon other mechanisms (e.g. CTLA-4, IL-10, TGFβ) dispensable for in vitro function (Sojka, et al. Eur J Immunol. 2009 Jun; 39(6): 1544–1551. PMID: 19462377; Kataoka et al. International Immunology, Vol. 17, No. 4, pp. 421–427; Read et al. J Immunol. 2006 Oct 1; 177(7): 4376–4383.) and probably other, less well-characterized mechanisms supporting FOXP3 expression and the Treg phenotype under an the no doubt complex set of environmental stresses encountered in vivo, but not in vitro.

It is likely that factors such as TRAF6 are necessary to maintain optimal FOXP3 expression, function and localization in the face of potentially diverse stresses in vivo. These can include abundant inflammatory cytokines and microbial products or reduced levels of Treg-supporting cytokines such as IL-2 and nutrients. Such stressors are not likely as prevalent in the more defined in vitro cell culture setting. Thus even unstable Tregs (such as those lacking

TRAF6) may be able to retain sufficient FOXP3 levels and sufficient adherence to the characteristic gene expression profile needed to allow for the deployment of some Treg suppressive mechanisms.

Concerning the assertion that the mechanism we describe is Treg intrinsic, we provide data in both the original and updated manuscript suggesting that, in fact, our mechanism is, at least in part, sensitive to environmental inputs. Specifically, we find that TRAF6 deficiency exacerbates the loss of FOXP3 levels in Tregs caused by pro-inflammatory cues (IL-6, LPS; see Figure EV1). These observations are in accord with those previously reported by Muto et al. who showed that Treg-destabilizing lymphopenic in vivo conditions and the presence of inflammation were able to enhance FOXP3 down-regulation and the generation of “exTregs” in the absence of TRAF6 (Muto et al. PlosOne 2013. Taken with our prior findings (Chen et al. Immunity 2013), which clearly implicated proinflammatory cues as capable of affecting ubiquitin-driven regulation of FOXP3, these observations suggest that non-intrinsic elements are at play.

We now include a discussion of the in vivo-in vitro disparity of our findings relating them to the broader field of Treg stability literature and speculate on the potential responsiveness of TRAF6-mediated Treg support to environmental inputs.

- The immunofluorescence microscopy data shown in figure f and supplementary figure 6 is not of sufficiently high quality to justify the authors conclusions. Higher magnification confocal microscopy images should be included to more convincingly demonstrate the altered localisation of FOXP3 in Traf6^{-/-} T regs.

We thank the Reviewer for their suggestion. To address this concern we have integrated confocal microscopy images that clearly visualize a disrupted cellular localization of FOXP3 in TRAF6-deficient iTregs. Images were obtained with a confocal microscopy EZ-C1 (available through the SKCC cell Imaging Facility) with FIJI-IMAGE J software. High magnification images from this analysis are now included as Fig EV5B in the revised manuscript. These results, along with corroborating Imagestream data (see Fig. 5E) support our assertion that without this E3 ligase, FOXP3’s expected nuclear localization is perturbed.

- The authors should clarify if the WT controls indicated in Figure 2, figure 5 and figure Sf6 are wild type C57/B6, or Foxp3 Cre mice which are more appropriate controls.

We thank the Reviewer for pointing this out. In the indicated experiments (and unless otherwise noted) wild type controls were either Traf6^{fl/fl}, Foxp3^{Cre}- littermates or Traf6^{wt/wt}, Foxp3^{Cre+} mice. In our revised manuscript we have made efforts to indicate in the figure legend and revised figure labels the exact genotype of “wild type, WT” controls used.

- The data shown in Sfig 7 suggests that FOXP3 K262 mutation reduces FOXP3 stability in vivo. It is not clear how this fits with the proposed mechanism of TRAF6 mediated K63 ubiquitination which should alter subcellular localisation rather than stability.

We have added additional results suggesting that TRAF6 expression and K63-Ubiquitination indeed influence not only FOXP3 localization, but also the stability of the FOXP3 protein pool. This data is now included as Supplemental Appendix Figure S3. We believe that this added dimension of the data set now presents a more complete picture of the potential link between TRAF6-mediated K63 modification and FOXP3 protein stability in vivo and in vitro. At present the precise mechanistic connection between a defective TRAF6-mediated K63-ubiq. at residue K262, perinuclear accumulation of FOXP3, and protein instability remains unknown. While this is an ongoing research topic, we suggest that degradative processes may be enhanced and stabilizing modification impaired in the absence of TRAF6-mediated K63-ubiquitination, and we discuss some possible scenarios in our revised manuscript.

Minor points

- To avoid confusion the authors should use the correct nomenclature when referring to FOXP3 and TRAF6 proteins and Foxp3 and Traf6 (in italics) genes, particularly since the study incorporates measures of both.

We thank the Reviewer for pointing this out. In our revised paper we have made efforts to use the correct nomenclature for genes and proteins.

- No information on antibody sources and clones used is provided in the methods section. Overall the methods section lacks the appropriate level of detail required to allow replication of the experiments described.

We apologize for the lack of detail in our original manuscript. We have now added a table of key antibody information to our supplementary information (Appendix Table S1) and we have added more experimental detail to our methods section and figure legends as well.

- There are a number of typographical errors in the labelling of figures which may not be picked up by copy editors.

Again, we apologize for these unfortunate errors. Extra care has been taken to correct them in the revision of our manuscript.

Referee #2:

Ni et al studied the functions of TRAF6 deletion in Foxp3+ Treg cells. TRAF6 deletion impaired Treg function, and TRAF6 promoted K63 ubiquitination at lysine 262. The latter event was shown to be required for optimal Foxp3 function.

Main comment

There is already quite a literature on TRAF6 being required for optimal Treg function, including references 17-23 cited in the paper and others, and so the new part of the data is that relating to K63 ubiquitination of Foxp3. I.e. Given prior studies of the effects of conditional deletion of TRAF6 in Foxp3+ Tregs I don't think Figures 1 and 2 add much to the paper, and these Figures could well be deleted or simply consigned to the supplement. In contrast, the new material relating to the K63 ubiquitination of Foxp3 is of significant interest.

We thank the Reviewer for expressing their interest in our novel findings. While we agree that much of the contents of Fig. 1 serves to confirm prior work by other groups, moving it and Fig. 2 from the main body figures to the Supplemental Materials would likely impact the flow of the data presentation negatively without considerable benefit. It should also be pointed out that our finding in the B16 tumor model have, to our knowledge, not been reported to date (much of Fig. 2). Thus, to help maintain a logical presentation of results while emphasizing both novel findings and those serving an important validation function, we respectfully choose to leave Fig. 1 and 2 in the main figures.

Minor comments

1. As it stands, the authors should document the purity of the cell populations under study in Figure 1; i.e. iTreg development is never 100% so what proportion of cells were Foxp3+ in panel a, and likewise, what proportion of the cells in panel b were Foxp3+?

This is an important point. We now provide a flow cytometric confirmation of the FOXP3 up-regulation typically seen in our murine iTreg-skewing experiments including those depicted in Fig. 1A. We also supply qRT-PCR measurements of key lineage-identifying factors in the other CD4+ T cell subsets generated in vitro. A representative gating strategy and intracellular FOXP3 staining of the human Tregs and non-Treg CD4+ T cells sorted from healthy donor blood in original panel 1B are also provided (see Appendix Figure S1A and B).

2. In Fig. 3, panel B, what is shown in lane 3 (lysate?), and where is the expected reciprocal pull-down (IP HA and IB Flag)?

We apologize for the labeling error. In the original panel 3B, the 3rd lane depicts the whole cell lysate input for the co-IP experiment. We have repeated this experiment with a reciprocal pull

down of HA-labeled protein (FOXP3) with blotting for Flag (TRAF6) (data not shown), but we chose to include the results of an endogenous reciprocal co-IP of unlabeled TRAF6 and FOXP3 from iTregs in the data set (see revised Fig. 3D) as this approach more directly illustrates the interaction of the two proteins in a more relevant cell population.

3. Some of the text simply goes whacky in places, e.g. What to make of the last sentence of the first paragraph of the Discussion? Likewise, the sixth paragraph of the Discussion needs attention ("we recently suggested positive relationship between TRAF6 expression and Treg function").

We apologize for the grammar and typographical errors in our original version. In revising our manuscript we have made extensive grammar and language edits with the help of native English speakers. We hope the current version is much improved and clearly presents our findings.

Referee #3:

Overall this is an interesting study that increases our understanding of the posttranslational mechanisms that regulate Foxp3 activity and thus Treg function. Although the first part of the study is not novel as mice deficient for TRAF6 in Treg have been described before and TRAF6 was shown to be essential for Treg functional activity in vivo (ref 22), the data on TRAF6-mediated K63 ubiquitination of Foxp3 are novel. The authors show that TRAF6 is able to add K63-linked polyubiquitin to FoxP3 specifically at residue K262, and that this ubiquitination is required for nuclear translocation and FoxP3 function. The authors also nicely show that this ubiquitination event is important in vivo by a Teff transfer model for colitis, where overexpression of wild-type FoxP3 in the transferred T cells by lentiviral transduction induces protection, but the K262R mutant does not. The reduced Treg functionality in TRAF6-deficient Treg mice also resulted in greater immune responses and anti-cancer activity in a B16 tumor model. The authors suggest that targeting TRAF6 could have potential to break tolerance and increase anti-tumor responses. Some issues still need to be addressed.

Thank you for the accurate summary and the kind remarks expressing interest in our findings.

Major comments:

- Fig. 3: There are several issues with this figure.

1. the legend should better describe how the experiment was performed and what exactly is shown.

We apologize for the lack of clarity. We have rewritten the figure legend in question to be more descriptive adding more experimental details, and we have corrected some labeling issues as well.

2. How can the authors exclude that the ubiquitination signal in their Co-IP experiments is coming from FOXP3 and not from another co-IPed protein.

In our co-IP experiments, we treated our samples (after pulldown) with conditions intended to denature and generally dissociate interacting proteins. As stated in response to a similar concern raised by Reviewer #1, our sample loading buffer for PAGE resolution of co-IP'ed sample and immunoblotting contained β -mercaptoethanol (14.3 M) and SDS (10%), and we boiled our samples in this buffer prior to running the SDS-PAGE gel (typically an SDS concentration of 20% was used in preparing gels). These denaturing conditions, reciprocal co-IP approaches, and the use of different cell lines and primary cells should have minimized the chances of mistakenly attributing the observed K63-type ubiquitin signal to another member of the FOXP3 protein complex.

3. Fig. 3D: labelling of the lower 2 panels is wrong. This experiment can also benefit from a similar set-up using TRAF6 KO cells as negative control to prove specificity of the antibodies used.

We thank the Reviewer for rightly pointing out the unfortunate labeling error. It has been corrected in the revised Fig. 3. We have not carried out an endogenous co-IP experiment using

TRAF6KO-derived iTregs as controls since use of primary cells in these repeat experiments can be prohibitive in terms of the cell numbers and donor mouse numbers required. Also, while this suggested control would be effective at uncovering a lack of specificity in the anti-TRAF6 antibody used in these experiments, we obtained it from a highly reputable vendor (Cell Signaling Technology), and we have not encountered such issues in our regular use of the reagent to date.

4. Why do the authors use ubiquitin mutants to make their point for K63-Ub in the case of Hek293T cells, while using K63-specific antibodies in the case of primary cells. K63-specific antibodies should also be used in case of the cell line.

Work with transfection-permissive cell lines lends itself more readily to approaches utilizing multiple expression constructs including ubiquitin mutants. This allowed us to exploit conditions where only specific modes of ubiquitination (i.e. K63 ubiquitination) would be present. For practical reasons, our work with primary cells relied more heavily on K63-specific antibodies as a means to visualize the abundance of certain modified FOXP3 species. In response to the Reviewer's suggestion, we now include data generated in a cell line using a method for specifically assessing K63-ubiq of Foxp3 similar to experiments done in primary cells with K63-specific antibodies. In this experiment, we used a K63-TUBE reagent (specifically recognizing K63 polyubiquitin chains) to pull down proteins labeled with this specific ubiquitin modification. The presence of FOXP3 within this K63-Ubiq labeled pool was then determined in the presence or absence of functional TRAF6 by immunoblotting (please see the revised Figure 3A in the revised manuscript). We also used this reagent for specific pulldown of K63-modified proteins from cell lines in the new Figure EV3E.

5. Why do the authors show acetylated FOXP3 (mentioned in legend, while no reference to K63-UB is made).

We apologize for the confusion created by our omitting important context concerning this observation. Please see our responses to Reviewer #1's comment on this point, but, in short, we have removed the figure in question from the manuscript. Our revised Discussion now speculates on the potential cross-talk between K63-ubiquitination and other modes of posttranslational modification (including acetylation) in the regulation of FOXP3 – a topic beyond the scope of the current study.

- The authors map TRAF6 binding to the zinc finger domain in FoxP3 (Figure S4), but do not show whether TRAF6 binding is required for FoxP3 ubiquitination (for example by comparing TRAF6-mediated ubiquitination of the FoxP3 mapping clones C2 vs C3 from figure S4).

The Reviewer brings up a valid point. To address this we now include data showing that FOXP3 K63 ubiquitination is only seen among the deletion variants that retain the ability to interact with TRAF6 (see Figure EV3E). As expected, Foxp3 mutants lacking the zinc finger and leucine zipper domains displayed reduced levels of this modification, confirming the importance of interaction with TRAF6 for this mode of post-translation modification. Unexpectedly, one TRAF6-interacting mutant was not found to be ubiquitinated ("C3"). As this variant lacked a proline rich domain, this may reflect proximity of the proposed target lysine or disruptive conformational changes that interfere with TRAF6-mediated modification, but not interaction. We mention these findings and possible explanations in our revised manuscript.

- The TRAF6-C70A point mutant (Fig 3A) causes more extensive structural changes to TRAF6 than only disrupting its E3 ligase activity. A good alternative for inactive TRAF6 could be the L74H, which disrupts E3 ligase activity (PMID: 28404732). Moreover, this published paper describing a ligase-inactive TRAF6 knock-in mouse also indicates ligase-independent roles of TRAF6 (which are also disrupted by C70A). The latter should be discussed by the authors in view of their own results and might be important for the interpretation of the Treg-specific TRAF6 KO data.

We thank the Reviewer for this insightful suggestion. We have incorporated the more specific ligase-defective mutant (L74H) into our experiments. As shown in the revised Fig. 3A, we find that this TRAF6 mutant is, like C70A, defective in executing K63 ubiquitination of FOXP3. Furthermore, reconstitution of primary murine CD4+ T cells lacking in TRAF6 (from Traf6 cKO

mice) failed to restore either FOXP3 K63 modification or activity in stark contrast to wild type TRAF6 (see Figure EV4C, D of our revised manuscript).

We now also discuss the possibility that ligase-independent activities of TRAF6 may be at play in Tregs with citation of the indicated reference.

- Could redundancy explain why TRAF6-deficient Tregs are possible to generate in vitro?

This is an excellent point. It seems likely that the initial activation of transcription the Foxp3 gene is depends less on TRAF6 than the maintenance of FOXP3 protein levels does. Others have found that TRAF6 can be dispensable for activation of NFkB downstream of T cell Receptor (TCR) stimulation (King et al. at Immunol 2006; 12:1088–1092.) with other ligases potentially contributing (Stempin et al. J Biol Chem 2011; 286:37147–37157). As the induction of Foxp3 expression in vitro and in vivo are heavily dependent on this pathway, a redundancy in TRAF6's contribution in this aspect of Treg biology seems likely. It is also worth noting that some critical determinants of stable Foxp3 transcription in established Tregs (e.g. CNS2) appear to be dispensable for initiating Foxp3 expression in the thymus (Fang et al. Cell. 2014 Aug 14; 158(4): 749–763. PMID: 25126783). Thus the differential requirement of key Treg-factors at discrete phases of Treg biology appears to be a running theme.

- Why do the authors use the B16 model which is known to be weakly immunogenic? Use of another more immunogenic mouse model seems more relevant to show the effect of Treg deficiency.

This is a good point. B16 melanoma model is widely used to study the regulation of anti-tumor immunity, and since the growth of these tumors is quite robust, sizable effects of treatment and genetic alterations on tumor progression can be very noteworthy. Also in this model tumor growth has been repeatedly shown to be greatly hampered by Treg depletion (demonstrating a key role for these cells in tumor progression in this model). Never the less, at the Reviewer's suggestion, we have repeated our tumor challenge experiments of Traf6^{fl/fl} Foxp3^{Cre+} mice and their wild type littermates using the more immunogenic MC38 colon cancer model to corroborate our findings and strengthen the data set. As with our earlier experiments in the B16 model, we find that Treg-restricted TRAF6 deficiency severely delays implanted tumor growth while enhancing anti-tumor immunity at the expense of Treg presence in the tumor microenvironment. These results are now included as new Figure EV2 in the revised manuscript.

- The writing style should be much improved and the manuscript needs to be revised by a native English speaking person.

We have made extensive edits to the manuscript with the help of native English speakers as suggested. Hopefully the current version reads much better.

Minor comments:

Abstract: rephrase the sentence "... perinuclear accumulation, disrupted function ..."

At the Reviewer's suggestion we have rephrased this sentence by adding some missing words.

Figure labelling in text should be corrected (Fig.Fig should be Fig.)

We apologize for this unfortunate redundancy in figure citation that arose in the formatting process. All instances of this error have been corrected in the revised manuscript.

Fig 1D : the truncated scale for the spleen cell count could lead to interpretations of a larger difference than there actually is.

This is a good point. We have changed the scale of the lower graph in panel D of the original Figure 1 to better reflect the relative effect size in spleen and lymph nodes.

Page 10: the text mentions effects of K63-Ub at K262 on FOXP3 stability, while the data only show effects on nuclear localisation

In the new Appendix Figure S3 we now include results that indicate the loss of TRAF6 expression or the K262 residue (K262R), the site of K63-Ubiquitination, results in accelerated FOXP3 protein turnover and a heightened level of K48-Ubiquitination (both indicative of an unstable protein pool, which is in line with in vivo observations). We have modified the statement in question to more accurately reflect our findings and we discuss the potential linkage between localization and stability more in the revised paper as well.

Page 12, discussion. Also the lineage tracing experiment that shows loss of FoxP3 expression in TRAF6-deficient Tregs (effectively reverting them back to Teff cells) should be mentioned here (PMID: 24058613). This should not be dependent on sampling time as mentioned in the current discussion.

The Reviewer rightly points out that the lineage tracing experiments of the indicated paper should be discussed here. While we do cite the paper by Muto et al. repeatedly in this portion of the Discussion, and we also mention their key finding that Tregs lacking TRAF6 are unstable, it is possible that we did not unambiguously relate this results to our own results adequately. We apologize, and the revised manuscript includes a revised Discussion of the experiment in question and the minor incongruity between our observations and those reported previously by Muto et al. Essentially we posit that taking into account the unstable nature of TRAF6-deficient Tregs that was clearly demonstrated by the fate-tracking experiments of Muto et al., it is possible that a number of factors may influence baseline FOXP3+ cell frequency in the tissues of TRAF6 knockout mice. These could include but are not limited to hard-to-define stresses or microbes in the animal environment (e.g. differences in caretaking/housing, animal facility conditions, microbiome), which may influence the intricacies and severity of phenotype seen in these mice which undeniably display issues with immune system control. Importantly, it should be pointed out that despite differences in observations of basal Treg frequencies, both our study and that of Muto et al. are in agreement in regards to the profoundly impaired in vivo function and stability of Tregs lacking TRAF6 activity.

The authors make the statement that potential therapies targeting TRAF6 may be highly effective at breaking tolerance. This is an overstatement, knowing the many other functions of TRAF6.

The Reviewer's point is well taken. We have modified statements made on pages 10 and 14 of our original manuscript specifically to qualify our perhaps over-enthusiastic assessment of the current study's implications that require additional in-depth investigation to realize. We also call attention to the other functions of TRAF6 that may complicate attempts to target the enzyme in the cancer setting.

Reference 22 should be corrected (show full name of last author)

We apologize for the unintended shortening of this and all other references in our original manuscript. We have reformatted the references in our revised paper in accord with journal requirements showing names of the 1st 20 authors of each paper in the Reference List. We reformatted references to match Embo J requirements both in line (i.e. author name and date) and in the citation list (i.e. citations are listed alphabetically with the names of the first 20 authors listed and the name of each journal abbreviated and italicized).

Details and source of plasmids + antibodies used should be provided.

We now include the sources of plasmids used in these studies in the Methods section, and a table listing the antibody clones and their source vendors has been added to the Supplemental Material (see Appendix Table S1).

Thank you for submitting a revised version of your manuscript. Please accept my apologies for the delay in getting back to you, owing to the seasonal holidays. Your study has now been seen by the original referees whose comments are shown below.

As you will see they find that all criticisms have been sufficiently addressed and recommend the manuscript for publication. However, before we can officially accept the manuscript, there are a few editorial issues concerning text and figures that I need you to address.

Referee #1:

The authors have satisfactorily addressed all points raised by this reviewer.

Referee #2:

The revised paper has alleviated my concerns and is a respectable effort.

Referee #3:

The authors have done a very good job in addressing all comments in detail. They also performed several extra experiments to document their answer. I have no further comments.

Corresponding Author Name: Ling Lu
Journal Submitted to: The EMBO journal
Manuscript Number: EMBOJ-2018-99766R